# Wavefront Shaping of Scattering Forces Enhances Optical Trapping of Levitated Nanoparticles

Mélissa Kleine [1], Michael Horodynski [2], Stefan Rotter [3], Yacine Amarouchene [1], Yann Louyer[1], Mathias Perrin [1] & Nicolas Bachelard [1] ✉

Optically-levitated nanoparticles in vacuum offer a pristine platform for high-quality mechanical oscillators, enabling a wide range of precision measurements and quantum technologies. A key performance metric in such systems is the stiffness of the optical trap, which is typically enhanced by increasing laser power—at the cost of unwanted heating, reduced coherence, and enhanced quantum backaction. Here, we demonstrate an innovative route to increasing trap stiffness: wavefront shaping of the optical field. By tailoring the spatial phase profile of the trapping beam, we significantly boost the mechanical confinement of subwavelength particles without raising the optical intensity. Remarkably, this enhancement arises from a selective reduction of non-conservative optical forces, while preserving the conservative restoring forces that define trap stiffness. As a result, mechanical nonlinearities are also reduced, improving stability at low pressures. Our findings challenge the long-standing assumption that diffraction-limited focusing is optimal for dipolar Rayleigh particles, and establish wavefront shaping as a powerful, readily applicable tool to control optomechanical forces in levitation experiments. This opens avenues for minimizing backaction, reducing thermal decoherence, and expanding the range of materials that can be stably levitated.

Optical levitation offers the tantalizing prospect of harnessing quantum effects at the mesoscale for both fundamental as well as technological purposes, and multiple milestones have recently been reached in the pursuit of these goals[1]. Through the focusing of a laser beam, particles with sizes in the nanometer or micrometer range can be trapped by an optical tweezer in high vacuum, resulting in under-damped oscillations around an equilibrium position. In contrast to other optomechanical devices[2], the spring constant (i.e., stiffness) that governs the oscillator dynamics scales here linearly with laser intensity, providing remarkable tunability. Large stiffnesses are, in particular, required to overcome decoherence rates and perform ground-state cooling of nanoparticles[3–6] as well as to achieve high-

performance sensors[7]. Importantly, trapping has to be photon-efficient—i.e., provide maximal stiffness for minimal optical intensity, as levitated objects in high vacuum are known to heat up substantially due to residual absorption, which is detrimental for quantum coherence[8] and can lead to particle melting[9]. Operating at reduced power is also of major importance to mitigate photonic recoil that hampers cooling[10]. Several theoretical proposals have therefore been put forward to increase a trap's stiffness at fixed or reduced intensity, for example, by using different materials and/or particles' dimensions[11,12]. Yet, no experimental realization of these concepts with optically levitated nano-objects in high vacuum has been reported so far.

[1]CNRS, Université de Bordeaux, LOMA, UMR 5798 Talence, France. [2]Department of Physics, Massachusetts Institute of Technology, Cambridge, MA 02139, USA. [3]Institute for Theoretical Physics, Vienna University of Technology (TU Wien), A–1040 Vienna, Austria. ✉e-mail: nicolas.bachelard@u-bordeaux.fr

In practice, the optomechanical stiffness is governed by the interplay between the different optical forces produced by the trapping field, which divide into conservative (gradient) and non-conservative (scattering) components[13]. Gradient forces attract the particle towards intensity maxima, whereas scattering forces commonly push it away along the optical axis, thus weakening optical confinement. For microscale particles (i.e., dimensions largely exceeding the wavelength), which exhibit numerous Mie resonances[14], stiffness is strongly hampered by significant scattering forces, whose contributions can be easily tailored—e.g., to balance gravity[15]. In contrast, levitated subwavelength nanoparticles are often approximated as simple Rayleigh scatterers (or point-like dipoles) with negligible mass, for which gradient forces dominate and lead to increased stiffnesses. Nevertheless, due to an inherently complex dipolar polarizability, the scattering force remains nonzero, shifting the equilibrium position away from focus[16], where field intensity—and therefore stiffness—is reduced. It has long been assumed that, unlike for microparticles, the relative contributions of gradient and scattering forces in a diffraction-limited laser field provide optimal confinement of nanoparticles[17], thus leaving no straightforward route for improvement.

In domains like multiple-scattering media[18–20] or complex beam generation[21], wavefront-shaping approaches, relying on devices such as Spatial Light Modulators (SLM), have been tailored to control optical-field distributions. In particular, in the case of single microparticles, a spatially-modulated wavefront can reorganize the contributions of Mie resonances, thus reshaping light-matter interactions[22]. Notably, for large microparticles trapped in liquid—where motion is overdamped—spatially modulating the trapping field's phase has successfully been used to enhance optomechanical stiffness[23,24]. However, this strategy might prove ineffective for nanoparticles levitated in vacuum, which, unlike microparticles trapped in water, are expected to behave as point-like Rayleigh scatterers, displaying underdamped motion on very short timescales. Given these constraints, it remains unclear whether wavefront shaping can provide any meaningful control over these particles' optomechanical confinement.

In this work, we challenge and overturn this prevailing assumption by demonstrating that optomechanical stiffness in nanoparticle levitation can indeed be enhanced through wavefront shaping. We achieve this by spatially modulating the trapping beam's wavefront to reduce scattering forces while preserving gradient forces. Specifically, we implement an iterative optimization routine that shapes the incoming beam onto a focusing objective to maximize the stiffness of a subwavelength silica bead, thereby making the optical tweezer more photon-efficient. Numerical simulations confirm that this optimization reshapes spatially the scattering force, shifting the equilibrium position closer to focus, where the particle experiences stronger confinement. Experimentally, we validate this effect by analyzing modifications within the stochastic toroidal motions embedded into the particle's dynamics, revealing direct signatures of reduced non-conservative forces. Crucially, our findings demonstrate that acting on scattering forces—previously considered a fixed constraint—provides a viable pathway to enhance optical confinement even at the nanoscale. Importantly, as optomechanical confinement is enhanced and the particle moves closer to focus, we also observe a reduction in motional nonlinearities that influence the overdamped dynamics at low pressures.

## Results

### Experimental optimization

We experimentally implement a wavefront-shaping approach that restructures the focused trapping field in order to increase the stiffness of deeply-subwavelength levitated particles. As displayed in Fig. 1a, we levitate individual nanoparticles (radius $R$ ranging from 75 to 125 nm, red dot) inside a vacuum chamber (Supplementary Fig. S1) using an objective lens ($NA = 0.8$), which focuses an incoming laser (wavelength $\lambda = 1064$ nm, intensity $\approx 300$ mW, red beam) whose phase is modulated through a phase-only SLM ("Methods"). The SLM encodes non-uniform wavefronts, $\phi(r)$, which reshape the focused field while preserving its overall laser intensity. Below a pressure of a few mbar, the particle experiences underdamped oscillations along $x, y$, and $z$, which materialize as mechanical resonances in the power spectral densities (PSDs), $S_{jj,j\in[x,y,z]}$, emerging for a uniform (i.e., unmodulated) wavefront at around $100, 120$, and $30$ kHz for $x$, $y$, and $z$, respectively (Supplementary Fig. S2). The corresponding stiffnesses $\kappa_x$, $\kappa_y$ and $\kappa_z$ are readily deduced from their respective resonance frequencies. The black curve in Fig. 1b displays the axial PSD, $S_{zz}$, of a 125 nm-radius silica particle levitated using a uniform wavefront (black circle, inset). The beam's phase is then iteratively shaped through an optimization routine and converges towards an "optimized wavefront" (white-to-black concentric circles, inset), which simultaneously improves the axial frequency resonance (along $z$, red curve) and the transverse ones (along $x$ and $y$, Supplementary Fig. S5). The corresponding iterative enhancements in optical confinements performed throughout the optimization are reported in Fig. 1c. At each iteration, the ratios of the modulated stiffness $\kappa_{opt, i\in[x,y,z]}$ over their initial (i.e., uniform) values $\kappa_{0, i\in[x,y,z]}$ are displayed and converge towards final magnifications of 1.4, 1.5 and 2.5 along $x, y$, and $z$, respectively. When applied to nanoparticles of different sizes, our approach systematically enhances the stiffness (colored curves, Fig. 1d), requiring different optimized wavefronts (white-to-black insets). While the transverse stiffness enhancements follow similar trends when the size increases (yellow and green curves), the axial enhancement turns out to be systematically larger (blue curve). Interestingly, larger particles—bestowed with larger scattering cross sections—display stronger stiffness enhancements along all directions. This emphasizes that the optimization is more efficient on objects that are subject to stronger scattering forces. Remarkably, the wavefronts are customized to match the particle they have been optimized for and appear only partially efficient when applied to particles of different sizes (Supplementary Fig. S10). Our optimization is implemented through a gradient-free routine, which maximizes a cost function assembled from stiffness ratios and can be adapted to promote specific resonance directions (Supplementary Fig. S6). The wavefronts are expanded over a Zernike-polynomial basis, in which axis-symmetric and asymmetric elements appear to have different contributions (Supplementary Figs. S3 and S7). The routine proves very robust: Even though it can converge towards different final solutions associated with different optimized wavefronts, we constantly report comparable enhancements for a given cost function. Inherent to digital SLMs, flicker noise materializes here into harmonics with marginal contributions and spectrally bounded below a few kHz (Supplementary Fig. S9). This technical noise does not overlap with the mechanical resonances and, moreover, can be removed using analog SLMs[25].

### Numerical model

Simulations emphasize that, along the optical axis, the optimization effectively brings the equilibrium position closer to the focus where optical intensity is larger and optomechanical confinement is enhanced. The optimization performed in Fig. 1b, c is emulated through a numerical model, which computes, for arbitrary wavefronts, the exact optical forces' landscapes through an accurate expansion onto electric and magnetic multipoles[14] ("Methods" and Supplementary Fig. S11). The black curves on Fig. 2a, b display respectively the total axial force, $F_Z$, exerted onto a 125 nm-radius particle by a uniform and an optimized wavefront (insets), when its position is varied along the optical axis, $z$. Equilibrium locations ($z_{eq}$, vertical gray dashed line) coincide with the positions where the force cancels ($F_Z(z_{eq}) = 0$) and around which the particle oscillates following a dynamic governed by

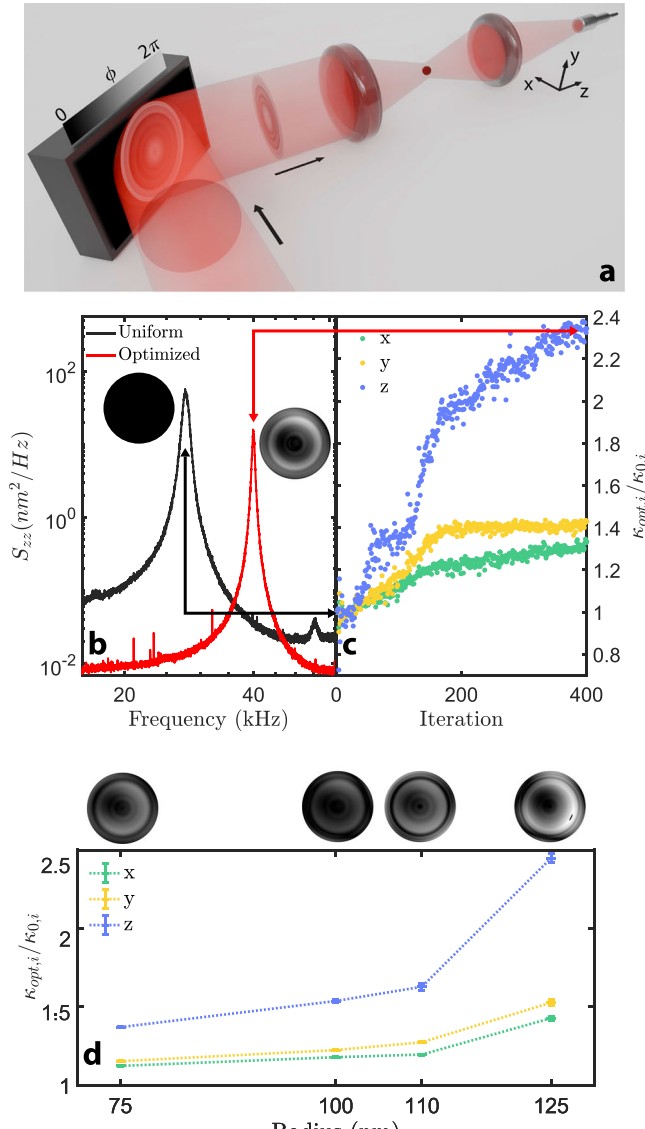

**Fig. 1 | Experimental optimization. a** A laser (red beam) is focused by a high-NA objective to levitate nanoparticles (red dot) in a vacuum. An SLM (black box) phase-modulates the beam and imprints a non-uniform wavefront (white-to-black concentric circles, $\phi(r)$ with $r$ the spatial coordinate on the SLM) that reshapes the trapping field (i.e., focused field). The transmitted light is collected by a second lens and injected into a fiber bundle to read out the nanoparticle's motion along $x$, $y$, and $z$. **b** PSDs $S_{zz}$ of a 125 nm-radius nanoparticle, measured for a uniform (black curve, black circle in inset) and an optimized wavefront (red curve, black-to-white circles in inset). **c** Evolution of the stiffness ratios $\kappa_{opt,x}/\kappa_{0,x}$ (green), $\kappa_{opt,y}/\kappa_{0,y}$ (yellow) and $\kappa_{opt,z}/\kappa_{0,z}$ (blue) throughout the optimization procedure. The iterations used in **b** are marked by the black and red double-arrow lines. **d** Optimized stiffness ratios obtained after applying the routine to nanoparticles of different radii (same color code as in **c**). The corresponding optimized wavefronts are reported in the insets. Error bars characterize the fluctuations of the final stiffness ratios under their respective optimized wavefronts.

the trap stiffness $\kappa_z = -d_z F_z(z_{eq})$ (gray dotted-dashed line). As expected, for a uniform wavefront, the scattering force pushes the nanoparticle away from focus to an equilibrium position $z_{eq} \approx 2.0 \, \mu m$. We observe that the optimized wavefront reshapes the axial force's landscape, which brings the particle closer to focus at $z_{eq} \approx 1.6 \, \mu m$, where the stiffness is increased by roughly a factor of 2.2 with respect to the uniform case. In line with experimental observations, simulations show that the trap is also more photon-efficient as the optimized wavefront can produce the same value of $\kappa_z$ as in the uniform case but

with now half the intensity (Supplementary Figs. S8 and S13). Once the particle lies closer to focus, we also improve the stiffnesses $\kappa_x$ and $\kappa_y$ along the transverse directions $x$ and $y$, respectively (Supplementary Fig. S12). Our model also highlights that, even if the nanoparticle is electric–dipole dominated (Mie parameter $2\pi R/\lambda \approx 0.74$), electric and magnetic multipoles do play a role[26]. The forces $F_z$ displayed in Fig. 2a, b are expanded into their main multipolar contributions, corresponding to electric-dipole ($F_{z,ED}$, blue), magnetic-dipole ($F_{z,MD}$, red) and electric-quadrupole ($F_{z,EQ}$, orange) terms. Yet, while we observe that higher terms are required to stabilize trapping in the uniform case of Fig. 2a, they have minor contributions to the mechanical stiffness that is mainly supported by the electric dipole (i.e., larger slope). As a result, the optimized wavefront mainly reshapes the electric-dipole landscape to improve the optomechanical stiffness $\kappa_z$, leaving higher terms unaffected (Fig. 2b).

Our numerical model also emphasizes that, as sketched in Fig. 2c, the optimized wavefront improves the stiffness by preserving the conservative gradient force while simultaneously strongly reshaping the non-conservative scattering force's landscape. Along the optical axis, a uniform wavefront (black inset) generates an almost Gaussian-like trapping potential ($U_0$, black line), in which scattering forces lead to the nanoparticle (gray dot) being trapped at a location $z_{eq}$ away from focus, where confinement softens. The optimized wavefront (white-to-black inset) barely modifies the potential ($U_{opt}$, red) while reducing the scattering. Thus, the nanoparticle is brought closer to focus, at a location where the potential quickly stiffens. This mechanism is illustrated in Fig. 2d, where the total optical forces $F_Z$ computed in Fig. 2a, b are decomposed into their gradient ($F_{g,0} = -d_z U_0$ and $F_{g,opt} = -d_z U_{opt}$, dashed lines) and scattering components ($F_{s,0}$ and $F_{s,opt}$, solid lines), displayed in black and red for a uniform and an optimized wavefront, respectively. We observe that the gradient-force's landscape (i.e., potential's derivative) remains largely unaffected by the optimization, while the scattering force is strongly shifted towards focus. This enforces a new equilibrium position, $z_{eq}$, where the slope of the gradient force significantly contributes to increasing the total stiffness. Alternatively, this stiffness improvement also reveals when considering the "effective" potential $U(z) - \int_0^z F_s(u)du$, whose confinement is substantially enhanced when the optimized profile is applied (Supplementary Fig. S14).

## Scattering-force modulation

A stochastic analysis of the nanoparticle's non-equilibrium dynamics confirms experimentally that wavefront shaping achieves a significant modulation of the scattering forces. Due to their non-conservative nature, scattering forces generate toroidal stochastic motions, which extend over distances much smaller than the particle's radius[16,27–30]. Yet, these motions materialize, both on the position and velocity probability distributions, as probability currents describing Brownian vortices (Supplementary Fig. S16). Denoting by $v_\rho$ and $v_z$ the norms of the transverse and axial velocities of a 125 nm-radius nanoparticle, Fig. 3a displays a histogram of the probability distribution ($P_v$, colored map) together with its current ($\bar{J}_v(v_\rho, v_z)$, black arrows), which are assembled from a 10 s-long trajectory obtained under a uniform wavefront (inset). Figure 3b reproduces the same analysis with the optimized wavefront of Fig. 1b (inset) and shows, as expected, confinement enhancements along both the transverse and axial directions, which are combined with a strong current (i.e., vortex) reduction indicative of a significant scattering-force adjustment. In Fig. 3c, the variance, $\langle \bar{J}_v^2 \rangle$, of the currents displayed in panels **a** and **b** is monitored when the pressure is varied, which emphasizes that this reduction remains when the dynamics of the particle varies. Yet, while current modifications are indicative of an adjustment in non-conservative forces, their interpretation remains qualitative (Supplementary Fig. S17).

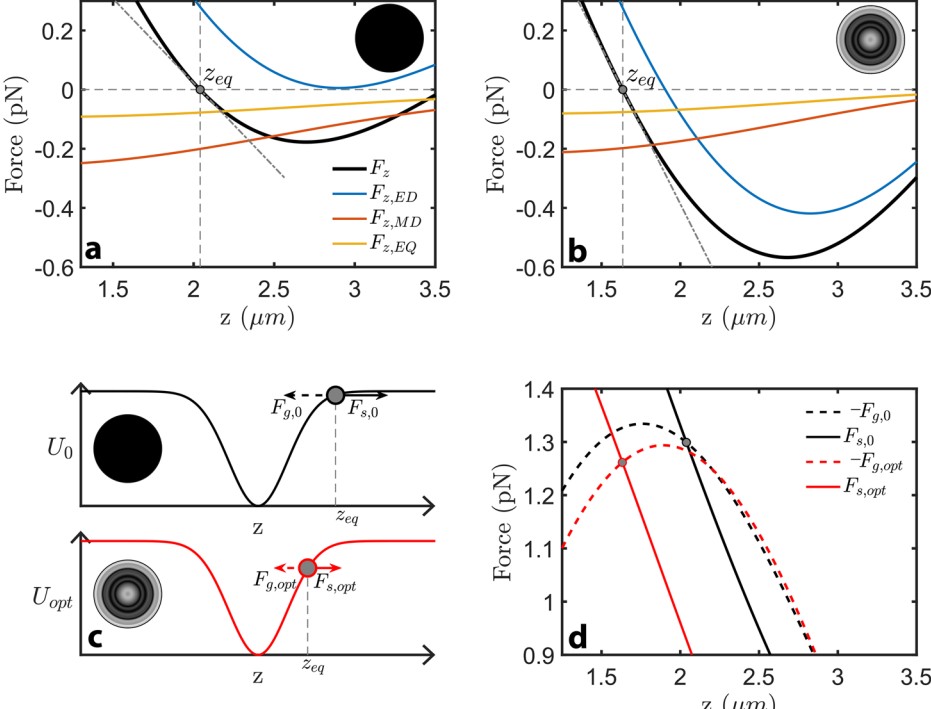

**Fig. 2 | Numerical simulations of forces' landscapes. a, b** Computed axial-force landscape ($F_z$, black) for a uniform and an optimized wavefront (insets) that are focused onto a 125 nm-radius nanoparticle. The forces are expanded into an electric-dipole ($F_{z,ED}$, blue), a magnetic-dipole ($F_{z,MD}$, red) and an electric-quadrupole ($F_{z,EQ}$, orange) contributions, which include interference terms (Supplementary Fig. S11) and fulfill $F_z = F_{z,ED} + F_{z,MD} + F_{z,EQ}$. Equilibrium positions (vertical gray dashed) fulfill $F_z(z_{eq}) = 0$, while the stiffness (gray dashed-dotted) is defined as $\kappa_z = -d_z F_z(z_{eq})$. **c** Along the optical axis, $z$, a uniform beam (black circle) produces an almost-Gaussian optical potential ($U_0$, black) that sets the gradient force landscape ($F_{g,0} = -d_z U_0$, dashed black arrow). The nanoparticle (gray bead) is trapped at an equilibrium position ($z_{eq}$) shifted away from focus, where $F_{g,0}$ balances the scattering force ($F_{s,0}$, solid black arrow). The optimized

wavefront (black-to-white concentric circles) barely modifies the potential ($U_{opt}$, red), while reducing the scattering force ($F_{s,opt}$, solid red arrow). The equilibrium is shifted to a location where the gradient force ($F_{g,opt} = -d_z U_{opt}$, dashed red arrow) matches $F_{s,opt}$ and the potential is significantly stiffer. **d** The axial-force distributions, $F_z$, displayed in **a** and **b** are decomposed into the sum of their respective gradient ($F_{g,0}, F_{g,opt}$), and scattering parts ($F_{s,0}, F_{s,opt}$), fulfilling $F_z = F_g + F_s$. The black dashed and solid curves represent respectively $-F_{g,0}$ and $F_{s,0}$ that are obtained under a uniform wavefront and which intersect at the equilibrium position $z_{eq} = 2\,\mu m$. The red dashed and solid curves represent respectively $-F_{g,opt}$ and $F_{s,opt}$ that are obtained under an optimized wavefront and which intersect at the equilibrium position $z_{eq} = 1.6\,\mu m$.

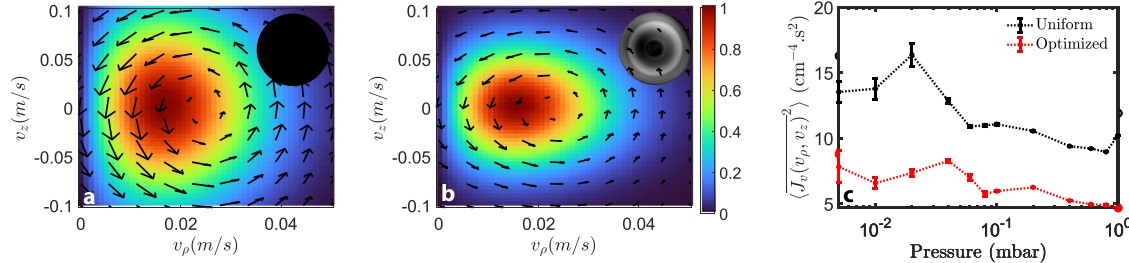

**Fig. 3 | Experimental observation of scattering-force modulation.**
**a, b** Probability distributions $P_v$ (normalized by their maximal values, colored map) and their respective currents $\bar{J}_v$ (black arrows) at 1 mbar in the velocity plane ($v_\rho, v_z$) for the uniform and optimized wavefronts of Fig. 1b, respectively (insets). **c** Averaged flux variance, $\langle \bar{J}_v^2 \rangle$, as a function of pressure for the uniform (black) and optimized wavefronts (red) of (**a**) and (**b**), respectively. The error bars characterize the standard deviations in $\langle \bar{J}_v^2 \rangle$ measured over multiple 300−ms-long time traces.

## Non-linearity reduction

At last, we observe experimentally that improving the stiffness also delays the emergence of Duffing non-linearities responsible for spectral deformations at low pressures[27,31]. When motional damping drops, thermalized particles start to explore the non-linear regions of the trapping landscape. Therefore, stronger confinements (constraining the oscillations' amplitude) and trapping the particle closer to the focus (i.e., further away from non-linearities) effectively reduce the influence of non-linearities (Supplementary Fig. S14). As Duffing

spectral deformations are more pronounced in the transverse plane, we plot in Fig. 4a the PSDs $S_{yy}$ for different pressures, measured for a 125 nm-radius nanoparticle, while applying the uniform (black) and optimized wavefronts (red) of Figs. 1 and 3. At a pressure of 10 mbar, the two PSDs are almost-Lorentzian, while below 0.1 mbar the one obtained under an optimized wavefront is clearly less subject to non-linearities. We show in the Supplementary Fig. S19 that central frequencies in the uniform and optimized cases are barely reduced when pressure drops. Interestingly, more quantitative insights on the

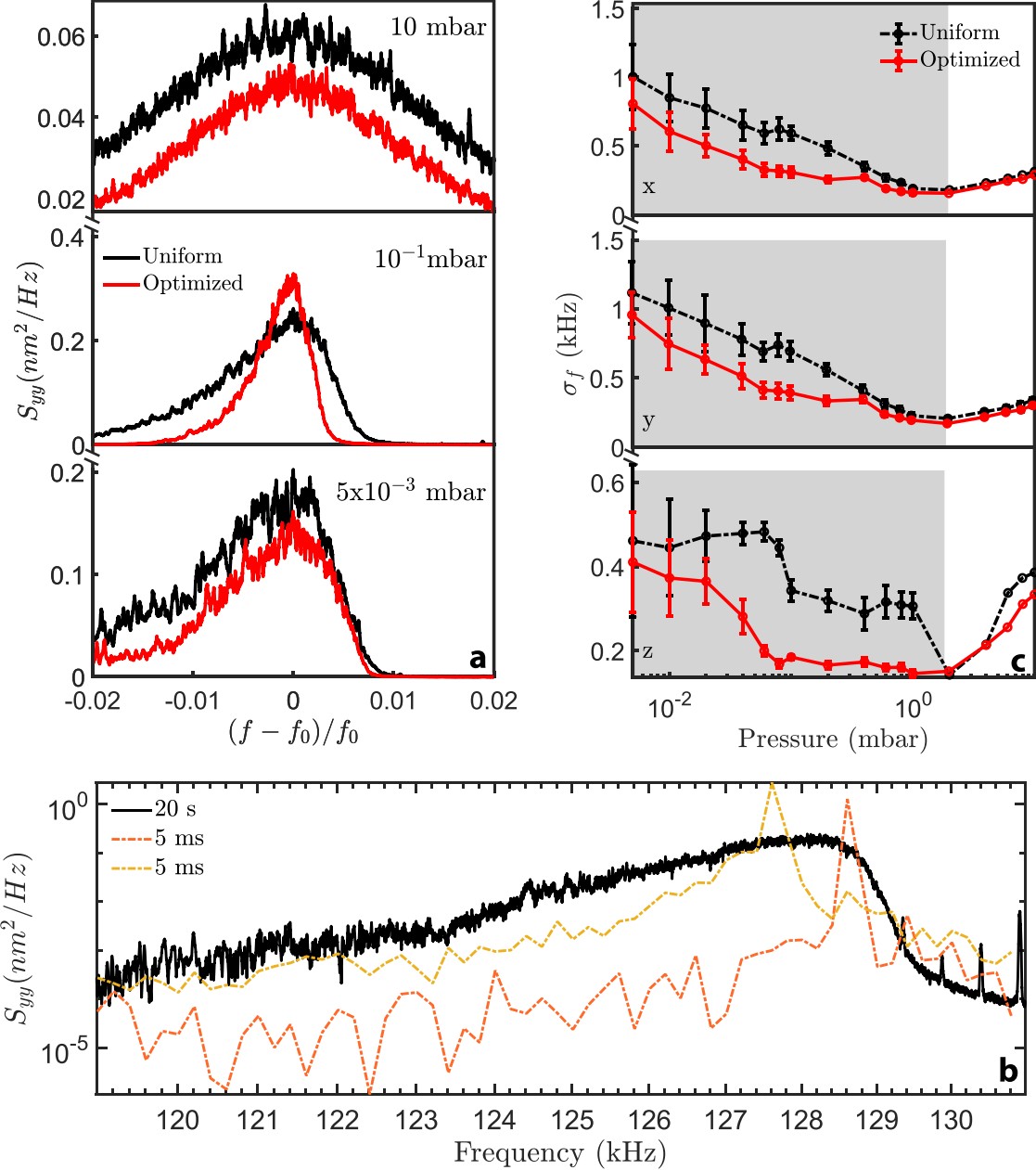

**Fig. 4 | Experimental reduction of non-linearities. a** PSDs $S_{yy}$ of a 125 nm-radius sphere measured at three different pressures under the uniform (black) and optimized (red) wavefronts of Fig. 1 and Fig. 3. The reduction of spectral broadening in the optimized case for pressures below 1 mbar emphasizes that nonlinearities are strongly reduced. **b** PSDs $S_{yy}$ obtained for a 20 s-long time trace (black) and two Fourier-limited 5 ms-long time traces (dotted red, dotted orange). **c** Standard deviations, $\sigma_f$, of 5 ms time-trace frequency distributions measured along $x$, $y$, and $z$ at different pressures. The black and red curves are obtained using the wavefronts of (**a**) (same color code). Above $\approx 2$ mbar, both curves display the same standard deviation. The gray regions indicate where $\sigma_f$ is lower (and non-linearities reduced) for the optimized wavefront compared to the uniform case. The error bars characterize the fluctuations in $\sigma_f$ measured over multiple 500 ms-long time traces.

reduction of non-linearities can be gained through short-term dynamics. For instance, at 0.1 mbar and under a uniform wavefront, Fig. 4b provides $S_{yy}$ assembled from 5 ms/20 s trajectories (orange/yellow dashed and black, respectively), which emphasizes that spectral broadening and the subsequent non-linear deformations originate from stochastic shifts of the short-time resonance frequency. In Fig. 4c, when pressure is varied, we plot the standard deviations of the short-time-resonance distributions along $x$, $y$ and $z$, which we assemble from frequency-distribution histograms (Supplementary Fig. S18). As expected, when applying the optimized wavefront, non-linearities—which are characterized by large deviations in short-time-resonance—are reduced below roughly 1 mbar (gray regions).

## Discussion

In conclusion, we have demonstrated that wavefront shaping can enhance the mechanical confinement of levitated nanoparticles by modulating non-conservative scattering forces. By leveraging commercially available spatial light modulators (SLMs), our approach can be seamlessly integrated into existing optical levitation setups, enabling more photon-efficient confinement. This allows for high trapping stiffness with minimal optical power, a crucial advantage in levitation experiments where photon efficiency directly impacts particle heating and backaction. While wavefront shaping has previously been explored for levitating single microparticles[32] or multiple microparticles[33,34], our findings underscore its potential for trapping

single nanoparticles—key to advancing quantum applications[1]. Building upon the remarkable progress recently achieved in wavefront shaping[35,36], we anticipate that this technique could be implemented non-iteratively[37] and transposed to different properties. For instance, it could help address major challenges in optical levitation, including stable trapping in high vacuum, reducing absorption losses and levitating particles composed of diverse materials[38], complex geometries[39] or embedded with emitters[40,41].

## Methods

### Experimental setup

Inside a vacuum chamber of about $5L$, the light produced by a linearly polarized, ultra-low-noise and water-cooled laser (Azurlight System, @1064 nm, 10 W) overfills the back aperture of an Olympus LMPlan IRx100 objective (NA = 0.8) with a filling factor of $f_0 = 1.1$. This forms an optical tweezer with a beam waist of 810 nm, in which silica nanoparticles (Microparticles GMBH) are trapped. The laser beam is modulated using a phase-only SLM (Holoeye, Pluto-2.1 NIR 149), whose zeroth diffraction order is spatially filtered using a 4f system. The SLM's flicker noise is characterized by maximal peak-to-peak phase fluctuations of 3%[42]. The light scattered from the particle is collected using a NA = 0.55 aspheric lens and injected onto a four-quadrant-like photodetector made of a 1–4 multimode fiber bundle (Thorlabs BF42HS01) coupled to two balanced photodiodes (Thorlabs PDB440C-AC). The signal is digitized using a 350 MHz-bandwidth DAQ (GaGe, TB3-RazorMax, 500 MS/s, 16 bits). The optimization is performed using a Nelder-Mead simplex algorithm, which corresponds to a gradient-free routine. The wavefronts are expanded along a basis made of 30 Zernike polynomials. The cost function is assembled from the ratios of the square of resonance frequencies obtained when applying the wavefront and in the absence of any modulation. A complete description of the procedure is provided in Supplementary Fig. S4.

### Numerical model

We compute the optical field focused onto the nanoparticle using a modified version of Debye's integral[13], in which the wavefront produced by the SLM is modeled through a thin-lens apodization function describing a non-uniform phase distribution. The scattered field is obtained using the Generalized Lorentz Mie Theory (GLMT), as an expansion on vectorial spherical harmonics. Integrating the Maxwell Stress Tensor over a spherical shell surrounding the nanoparticle, we obtain the exact expression of the total optical force landscape[43]. In parallel, the total force is also obtained through a computation of the electric and magnetic multipoles up to quadrupole contributions[44,45], which matches GLMT calculations with great precision (Supplementary Fig. S11). This multipolar approach gives analytical expressions of the force that reveal to be extremely useful to compute the conservative and non-conservative contribution[46] as well as to speed up the optical-force estimations required to emulate the experimental optimization procedure (Supplementary Fig. S15).

## Data availability

All the data that support the plots within this paper and other findings of this study are available from the corresponding author upon request.

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

## Acknowledgements

N. B. is grateful to L. Rondin for helpful discussions and dedicates this work to P. Bachelard for his wonderful support. M. P. thanks F. Gruy for discussions on the multimode expansion. The authors acknowledge support from the French Agence Nationale de la Recherche (GROOVE ANR-22-CE30-0016-01, QLeviO ANR-21-CE30-0006), the Région Nouvelle Aquitaine (UFOs, LeviTorques), as well as the French government in the framework of the University of Bordeaux's France 2030 programs GPR LIGHT and Quantum Matter Bordeaux. This research was also funded in part (M.H.) by the Austrian Science Fund (FWF 10.55776/J4729).

## Author contributions

M.K. performed the experiments, analyzed the data and contributed to writing the paper. M.H. provided preliminary data, while S.R., Y.A., and Y.L. provided scientific feedback and helped in the writing process. M.P. suggested the multipolar model and made all the numerical simulations. N.B. supervised the whole scientific process and wrote the main text as well as the Supplementary Information.

## Competing interests

The authors declare no competing interests.
