## [Transparent Peer Review file · Nature Communications]

Wavefront Shaping of Scattering Forces Enhances Optical Trapping of Levitated Nanoparticles

Corresponding Author: Dr Nicolas Bachelard

Version 0:

Reviewer comments:

Reviewer #1

(Remarks to the Author)

The manuscript "Wavefront Shaping of Scattering Forces Enhances Optical Trapping of Levitated Nanoparticles" by Melissa Kleine et al. Presents a study that deals with optimizing efficiency of optically levitated nanoparticle by the tailoring of the trapping beam phase by the SLM. The paper presents both the experimental data that demonstrate rather significant increase of the trap stiffness by a factor of almost 2.5. The experimental results are accompanied by the theoretical analysis of the trapping potentials and probability currents. In principle, similar types of trap stiffness optimizations had been performed before, however only for a particle trapped in the water and not for levitated one. In my opinion some of the current experiments with optically levitated particles would indeed benefit from increased trapping efficiency as suggested in the current manuscript.

The manuscript is clearly written and mostly quite easy to follow, methodology is sound. On the other hand, some technical details are missing and the limitations of the presented method should be discussed. The manuscript is accompanied by a detailed supplementary information which suffers from many formatting issues. And as the supplement does not go through the extensive editorial process similarly as the main text it has to improved.

In my opinion, there are two main issues that should be addressed in the manuscript (Supplementary information).

Firstly, the authors use the liquid crystal spatial light modulators (SLM) for optimization of the trapping efficiency. The SLMs introduce the flickering noise caused by the periodic refreshing the diffraction patterns. To my knowledge this is the case for the Meadowlark SLMs and I am not completely sure about Holoeye model used in this work. This intensity noise is present only in lower frequencies up to approximately 1 kHz, however its presence may be a limiting factor for some type of experiments with levitated particles. I think that it should be somehow mentioned in the text. In principle, it may be also beneficial to present the power spectral densities of the particle motion in much wider frequency interval, e.g. another panel in Fig S2 may show PSDs from 0 to few hundreds kHz.

Secondly, the choice of the selected Zernike polynomials used for trap efficiency optimization should be explained better. I guess that the axisymmetric ones are related to the trap efficiency increase while the Zernike polynomials having m index different from 0 actually compensate for potential experimental inaccuracies. In the case of axial Zernike polynomials (m=0) some connection to the theoretical analysis in SI 2.1 should be made, i.e. does Z^0_{38} resolve the narrowest rings considered in the optical force calculations? I can even think that one may perform the analysis based on the Fisher information how strongly each Zernike polynomial contributes to the final result (for future work). In the case of non-symmetric Zernike polynomials I would also guess that the ones with higher l and small m indices (e.g. $Z_{5^{+1}}$) may contribute to the optimal trapping phase.

Some further issues, comments and suggestions are mentioned below:

1. Page 2, sentence "However, this strategy is commonly believed to be ..." Is there any reference for this statement?
2. Page 6, sentence "Our model also highlights that, even if the nanoparticle behaves as a Rayleigh scatterer (Mie parameter $2\pi R/\lambda \approx 0.74$) ..." I think that it is quite established that for this size parameter the particle is not a Rayleigh scatterer. Actually according to Wiscombe Applied Optics 1980 one should use up to 5th order multipole expansion in order to correctly describe scattering of such a particle size.
3. Figure 2c. The illustration is a bit misleading in a sense that the equilibrium position is not in the minimum of potential. I completely understand that only the gradient force potential is plotted but I would recommend adding a curve corresponding

to scattering “potential.” The depiction would become cleaner

4. Figure 2 caption. Reference to supplementary figure S7. Is it indeed S7 or probably S10?

5. Page 8, sentence “... whose extension reveals much smaller than the particle size [16,24–27]” I think that a part of sentence is missing

6. Discussion on Page 9: is the trapping frequency or stiffness enhancement influenced by the changing pressure? I mean can the tiny changes of refractive index decrease the trap strength enhancement? Or could any changes to vacuum chamber induce an extra phase shift that would influence the enhancement.

7. Figure 4c, caption. It looks to me that σ_f is lower in the whole range of pressures, not just in the gray region

8. Supplementary Information: Please check the order of figures. Also correct the usage of commas around equations – on multiple occasions the paragraph after an equation starts with comma.

9. SI section 1.3. Could you provide more details about implementation of the optimization procedure. I.e. for given set of Zernike polynomials a particle trajectory of what length is measured (points, time), how is the trajectory processed to obtain the resonant frequency. What is the total duration of the optimization procedure. How many times is typically trajectory recorded (iteration count in `fminsearch` does not mean number of function calls). Could you also comment on the stability of the trapping setup during the optimization?

10. SI page 4, last paragraph: is X a vector? If so please use bold font

11. Figure S3d. I would find useful to visually separate the axisymmetric coefficient from the nonsymmetric ones (by a space or different background)

12. SI page 7, sentence “Nonetheless, the optimization is less efficient than when performed directly on the particle of the proper dimension.” Please use size or radius instead of dimension.

13. Figure S4: I would suggest to add data from figure 1 as well for direct comparison.

14. Figure S7ab. I would suggest to include dashed boxes that would show a magnified views in Fig. S7cd

15. SI page 9 paragraph 1. I think that it should be explicitly mentioned if the phase in each ring was constant or it also varied with polar angle which would probably introduce an equivalent of non-symmetric Zernike polynomials.

16. SI page 9 sentence “Zoomed-in views of both landscapes are provided in Figs. S7a and d.” Probably Figs 7c and d.

17. SI page 7 sentence “We also note that the coordinate z in the optimized case has been shifted such that the conservative part of the force (i.e., gradient force, see section 2.2) remains zero at $z = 0$ (i.e., the beam focus).” I think that this sentence may be a bit hard to understand and I would recommend reformulating it.

18. SI Equation (5): define H

Reviewer #2

(Remarks to the Author)

Kleine et al study the use of wavefront shaping to improve the mechanical properties of silica nanoparticles optically trapped in vacuum. A spatial light modulator (SLM) is used to generate trapping beams with tailored spatial phase profiles resulting in enhanced mechanical spring constants when compared to a diffraction-limited trapping potential. The authors then use numerical simulations to attribute this effect to the suppression of non-conservative optical forces, which shifts the equilibrium position of the nanoparticle closer to the trap focus. Finally, they report on the reduction of nonlinearities in the mechanical motion of the nanoparticle at lower pressures.

These results address key challenges in the fast-growing field of levitated optomechanics. Ground-state cooling of levitated mechanical systems has only been achieved in optical traps, where laser light is known to generate unwanted motional decoherence via scattering and absorption of photons. Wavefront shaping as demonstrated here could provide much-needed new tools to enhance trapping and suppress decoherence for quantum experiments with optically levitated nanoparticles. The paper also shows that nonlinearities can be suppressed using wavefront shaping, which could improve the stability of levitated oscillators in vacuum, another common problem with optical levitation.

The main claim of the paper, that wavefront shaping can enhance optical trapping of levitated nanoparticles, is justified and backed by experimental data (Fig 1). The paper is concise, well written and accessible to a broad readership. Prior work is appropriately acknowledged. I appreciate the careful analysis in the supplementary sections and the numerical simulations which provide detailed insights into the effect of tailored potential on the mechanics. However, the experimental plots lack error analysis (or mention of) throughout and there no strong theory plots to compare with the experimental datapoints. This makes all results (except Fig 1) less convincing.

This work presents the novel experimental tool of wavefront shaping for optomechanics with levitated nanoparticles, which has the potential to overcome key bottlenecks in the field. I believe is a worthwhile advance of relevance to the levitodynamics community. I would recommend publications in Nature Communications, but only if the following comments are addressed-

i. Based on Fig 1d, the authors state “This emphasizes that the optimization is more efficient on objects that are subject to stronger scattering forces”, indicating that their optimization works better with larger objects. Would this be the case even beyond the Rayleigh scattering regime? It would be useful to model the curves in Fig 1d to indicate the size range in which their optimization is beneficial and the maximum enhancement ratios that could be theoretically achieved.

ii. The authors refer to Fig S8 (numerical simulation) when claiming that “the optimized wavefront can produce the same value of kz as in the uniform case with half the intensity”. This should be straightforward to test experimentally and could strengthen the results by showing agreement with theory. Why has this not been done?

iii. Error bars are missing for all experimental plots (Fig 1d, Fig3c, Fig 4c), and no discussion of errors are provided. The strength of the conclusions is hard to judge without quantitative error bars.

iv. Fig 4c indicates that the nonlinearity is suppressed below 1 mbar. However, the suppression seems to get worse again below 0.01mbar across all three dimensions. Why does this happen? Given the simulations of the trapping potential and scattering forces, could a theoretical comparison be provided for the curves in 4c? It would be useful to know (even theoretically) if suppression can be achieved also at lower pressures commonly needed for most experiments in the field, particularly those motivated by quantum applications.

A few suggestions on terminology:

“the stiffness of the oscillator that governs its dynamics” – “the spring constant (stiffness) that governs the oscillator dynamics”

“magnifications” – “enhancement factors”

“whose extension reveals much smaller” – “whose extension is much smaller”

Reviewer #3

(Remarks to the Author)

Review of the manuscript: “Wavefront Shaping of Scattering Forces Enhances Optical Trapping of Levitated Nanoparticles” for publication in Nature Communications.

The manuscript by Kleine et al demonstrates a new iterative experimental method to enhance the optical trap confinement for dielectric spherical nanoparticles. It is based on SLM assisted wavefront shaping and consists in selective reduction of the scattering force while keeping the gradient force almost unaffected. As the result, the nanoparticle experiences a smaller axial shift away from the waist of the tightly focused trapping beam, which in turn leads to higher mechanical oscillation frequencies along all three directions with best enhancement achieved in the along the laser direction, corresponding to the lowest oscillation frequency. The reduction of the scattering force was demonstrated experimentally with the measurement of the joint probability distribution and current of the radial and axial particle velocities: in case of the optimized wavefront the magnitude of the vortex current in the phase space is smaller.

Finally, the optimized confinement was used to show reduction of the thermal nonlinearities experienced by the particle at lower pressures thanks to a better confinement of the particle motion within the harmonic region of the trap.

The nanoparticles used in this work are mostly Rayleigh scatterers with small deviations towards Mie scattering. This work goes beyond state of the art by showing that wavefront shaping can indeed outperform the diffraction-limited focusing, which was thought optimal Rayleigh particles. Therefore, the claims of the paper are convincing and extensively explored. The paper is written clearly and both experimental implementation and the experimental / numerical analysis are shown in a comprehensive way. The originality of the research approach is making use of wavefront shaping, commonly used for controlling light in complex media, for optimizing the control of nanoparticle motion. This will be of interest for the community exploring quantum optomechanics with trapped nanoparticles.

I would therefore definitely recommend the publication of this manuscript in Nature Communication under condition that the authors address the following minor issues that I found reading this manuscript:

1. For smaller nanoparticle radius the stiffness enhancement reduces. While the authors show that wavefront shaping mostly enhances the trapping due to the electric dipole force, I wonder whether the stiffness optimization would still work for perfect Rayleigh scatterers, for example single atoms in optical tweezers?
2. The scattering force is usually associated with the scattering cross section of a particle. In standard textbooks the scattering cross sections are usually defined for incident plane wave illumination, while the situation may be much different in case of a spatial phase modulated beam. Does wavefront shaping actually modify the effective scattering cross section of the particle? If yes, could it then also be used to modulate the absorption cross section in case of absorbing particles?
3. As far as I know, Silica is a nonmagnetic material and in the SupMat Section 1.1 I haven't found any information of the magnetic properties of the nanoparticles allowing to calculate the magnetic dipole polarizability. However, on Fig 2 as well as Fig S7, I see that the MD contribution seems important. Could the authors please provide more details on the magnetic dipole polarizability α_{MD} , used in the calculations?
4. The scattering force can be expressed as the product of the laser intensity and the gradient of the optical phase (see for example the equation 14.44 in the “Nano Optics” book by Lukas Novotny). On Fig. S8 c one can clearly see the reduction of the laser intensity along z between 0 and 3 μm , while the slope of intensity is not changed. This in turn reduces the scattering force while keeping the gradient force unchanged. But is there any additional contribution from the phase? Does wavefront shaping also affect the beam's wavefront at the trapping area in a way which could reduce the gradient of the phase?
5. The authors describe in details the procedure for the stiffness optimization in the SupMat Section 1.3, but I still miss the understanding of what is actually changed between each iteration: are the phase patterns corresponding to each Zernike modes tested one after the other at different contrasts?
6. Do the authors think it is possible to implement a non-iterative stiffness enhancement approach, similar to the Transmission Matrix measurement in static complex media (<https://link.aps.org/doi/10.1103/PhysRevLett.104.100601>), but based on the measurement of the particle's mechanical response to each applied mode?
7. The authors give the value of 430 nm for the beam waist which is roughly $0.404 \cdot 1064$ nm and the axial equilibrium position of $z_{eq}=2$ μm for a $R=125$ nm particle. These values seemed to me too small for the waist and too large (above the Rayleigh length) for the equilibrium position. By using the beam parameters: $NA=0.8$ (Numerical Aperture of the trapping lens), $f_0=1.1$ (overfilling defined as the ratio between the Gaussian beam waist before the lens and the lens radius) and the particle size $R=125$ nm, with help of the Optical Trapping Toolbox on MATLAB I found that a Gaussian waist $w_0=676$ nm

(which is $\sqrt{0.404} \cdot 1064$ nm) fits best the radial beam profile along x for a x-polarized incident beam (I used the definition: $I(x) = I_0 \exp(-2x^2/w_0^2)$). I also got the axial equilibrium position of $0.8 \cdot 1064$ nm which is approximately 850 nm. Could the authors please verify / justify the values presented in the manuscript?

8. On Fig. 3 c there are significant variations of the averaged flux variances but no errorbars. Would it be possible to discuss these variations or to provide estimated errorbars?

9. Just wanted to draw the attention of the authors to the fact that they give the value of 2200 kg/m³ for the mass density of Silica. It indeed agrees with the bulk density of Silica, but differs from the value of 1850 kg/m³ for the Silica nanoparticles provided by MicroParticles GmbH: <https://microparticles.de/en/properties>.

Version 1:

Reviewer comments:

Reviewer #1

(Remarks to the Author)

Dear Editor,

I have read the revised version of the manuscript by Melissa Kleine et al. with great interest. The authors have addressed all of my previous comments thoroughly and beyond my expectations, and I believe they have done the same for the other reviewers' remarks as well. In my earlier report, I stated that "the current experiments with optically levitated particles would indeed benefit from increased trapping efficiency as suggested in the present manuscript." I am now fully convinced that this is the case, and I recommend that the manuscript be accepted for publication in Nature Communications in its current form.

Reviewer #2

(Remarks to the Author)

The authors have provided a comprehensive response to my queries. They have shown experimental verification that power can be reduced (by almost a factor of 2) to achieve the same stiffness (addressing query 2), added appropriate error bars to their figures (addressing query 3), and provided discussions showing that their techniques are useful even at lower pressures (addressing query 4).

I do believe they misunderstood my first query about the effectiveness of their techniques for particle radius beyond a couple hundred nm. They describe that for the smaller particles that are typically trapped with 0.7 or 0.8 NA lenses (as is the case in their experimental setup), their optimisation shows improvement with size. However, the question was about larger particles that require lower NA trapping; does their optimisation still hold, and if not, is there an optimal size range where their techniques work?

I do not think statements such as "these simulations show that trapping becomes unstable when the radius exceed 140nm" and "Nonetheless, we are convinced that this technique can be straightforwardly transferred" are convincing (where in the simulations does the instability arise when they could change the NA in the simulations, and what convinces them that their techniques can be transferred?).

Nevertheless, this is perhaps a minor issue that simply needs clarification for completeness and readability. Overall, based on their modifications in response to referee feedback, the paper is much stronger now and I am happy to recommend publication.

Reviewer #3

(Remarks to the Author)

Wavefront Shaping of Scattering Forces Enhances Optical Trapping of Levitated Nanoparticles (NCOMMS-25-36074)

*First of all, we would like to thank the three referees for the time and effort spent on our manuscript. All the referees provided positive feedback combined with constructive remarks, which help to improve the quality of the paper. Referee #3 **accepted its publication with minor corrections**. Referees #1 and #2 provided very **positive comments, while asking to address specific points**. Specifically, Referee #1 highlighted that our work **could be beneficial to the levitation community** and stressed the **quality of our methodology** and writing. Referee #2 emphasized that this paper **addresses key challenges** in the field and also **mentioned positively our methodology**.*

*In this document, we address a point-by-point response to their reviews, for which **we acquired a new experimental data and run new simulations**. Moreover, as requested by Referee #1, we deeply reorganized the Supplementary Information. In particular, **we wrote 8 new sections, added 10 new figures and introduced a theoretical discussion**. We also adapted the main text and, as requested, we edited 3 figure panels. Alongside this reply, we provide edited version of the main text and the Supplementary Information, in which modifications are written in red and listed below for each comment.*

REVIEWER COMMENTS

Reviewer #1 (Remarks to the Author):

The manuscript “Wavefront Shaping of Scattering Forces Enhances Optical Trapping of Levitated Nanoparticles” by Melissa Kleine et al. Presents a study that deals with optimizing efficiency of optically levitated nanoparticle by the tailoring of the trapping beam phase by the SLM. The paper presents both the experimental data that demonstrate **rather significant increase of the trap stiffness by a factor of almost 2.5**. The experimental results are accompanied by the theoretical analysis of the trapping potentials and probability currents. In principle, similar types of trap stiffness optimizations had been performed before, however only for a particle trapped in the water and not for levitated one. **In my opinion some of the current experiments with optically levitated particles would indeed benefit from increased trapping efficiency as suggested in the current manuscript.**

The manuscript is clearly written and mostly quite easy to follow, methodology is sound. On the other hand, some technical details are missing and the limitations of the presented method should be discussed. The manuscript is accompanied by a detailed supplementary information which suffers from many formatting issues. And as the supplement does not go through the extensive editorial process similarly as the main text it has to improved.

We would like to thank Referee #1 for acknowledging the “benefit” our method could bring to the community and for stressing the quality of our manuscript and methodology. We also would like to thank Referee #1 for the quality and thoroughness of their comments, which we address point by point below and that helped to improve the quality of our paper.

Major comments:

- 1- Firstly, the authors use the liquid crystal spatial light modulators (SLM) for optimization of the trapping efficiency. The SLMs introduce the flickering noise caused by the periodic refreshing the diffraction patterns. To my knowledge this is the case for the Meadowlark SLMs and I am not completely sure about Holoeye model used in this work. This intensity noise is present only in lower frequencies up to approximately 1 kHz, however its presence may be a limiting factor for some type of experiments with levitated particles. I think that it should be somehow mentioned in the text. In principle, it may be also beneficial to present the power spectral densities of the particle motion in much wider frequency interval, e.g. another panel in Fig S2 may show PSDs from 0 to few hundreds kHz.

We thank Referee #1 for bringing this point to our attention. Flickering issues are known to be inherent to different SLMs’ technologies such as digital liquid-crystal SLMs (like the device used in the article) or Digital Micromirror Devices (DMDs). There, flickering originates from refreshing and the fact that each pixel is addressed using a binarized voltage [1].

*First, we would like to stress that **there already exist different SLM technologies, which are flicker-free and commercially available**. Here, one can mention analog liquid crystal SLMs, where pixels are addressed using analog voltage [2]. Such devices are commercialized by mainstream companies such as “Sintec Optronics” or “Holoeye” and provide extreme phase*

stability (i.e., no flicker). Moreover, the technology of Deformable Mirror is also an alternative that is inherently flicker-free and can now control hundreds of optical modes. Therefore, **our technique could be readily implemented in flicker-free conditions.**

Second, regarding the device used in this work (Holoeye, Pluto-2.1 NIR 149), its technology has been adapted to display reduced flicker. In particular, in reference [1] the authors investigated the flicker noise displayed by a Holoeye SLM relying on the same architecture. They reported a maximum peak-to-peak phase fluctuation of about 3%—similar to what is expected for our Pluto-2.1 NIR 149. Moreover, and as mentioned by Referee #1, to reduce flickering issues in digital SLM, constructors voluntarily reduce the refresh rate, which bounds spectrally the noise below a few kHz. This can be observed in Figure R1, where we plot (as requested) the unsmoothed spectra of Figure 1 of main text over a larger bandwidth. Up to 2 kHz, these spectra display flicker's peaks corresponding to 60 Hz harmonics (60 Hz being the display refresh rate). Even though there exist many peaks, they are ultimately spectrally narrow, such that their **cumulative contribution remains negligible compared to the informative signal** (i.e., the mechanical resonances). More importantly, we observe that the flicker harmonics (below 2 kHz) **do not overlap with our mechanical resonances** (above 20 kHz) **and thus have no influence onto them.**

As suggested by Referee #1, in some cases flicker noise can overlap with mechanical resonances. This would be the case for instance when considering micronsized particles, which display kHz and sub kHz frequencies [3]. As previously stated, flicker is rather reduced in our current digital SLM. Yet, if a lower noise level is required, one could reduce it using different techniques (e.g., cool the device [4]) or rely on a different technology (e.g., analog SLMs as discussed above).

Figure R1: **Flicker noise.** Unsmoothed power spectral densities recorded after the optimization introduced in Figure 1b and Figure 1c. The black and red curve are recorded for a uniform and optimized pattern, respectively. Below ≈ 2 kHz, both spectra display 60 Hz harmonics characteristic of flicker noise.

Related modification of the manuscript:

[1] Main text (page 4):

“Inherent to digital SLMs, flicker noise materializes here into harmonics with marginal contributions and spectrally bounded below a few *kHz* (Figure R1). This technical noise does not overlap with the mechanical resonances and, moreover, can be removed using analog SLMs [2].”

[2] Methods (page 12):

“The SLM’s flicker noise is characterized by maximal peak-to-peak phase fluctuations of 3% [1].”

[3] Supplementary Information: Figure R1 was added as the new Figure S8 and the new paragraph 2.4 was written.

- 2- The choice of the selected Zernike polynomials used for trap efficiency optimization should be explained better.

We thank Referee #1 for this comment and agree that such a discussion would be helpful for our readers.

I guess that the axisymmetric ones are related to the trap efficiency increase while the Zernike polynomials having *m* index different from 0 actually compensate for potential experimental inaccuracies. [...] In the case of non-symmetric Zernike polynomials I would also guess that the ones with higher *l* and small *m* indices (e.g. Z_{5}^{+1}) may contribute to the optimal trapping phase.

*First, we would like to stress that the relationship between the phase profile or wavefront (that is obtained as a linear superposition of the Zernike polynomials) and the trap stiffness is **very nonlinear**. Moreover, our procedure can converge towards different **local minima** (i.e., optimized wavefronts). As a result, the contributions of symmetric (Z_{2m}^0) and asymmetric Zernike polynomials ($Z_q^{p \neq 0}$) are **deeply intertwined and strongly influenced by one another**.*

*That being said, **we have collected new experimental data, which seem to confirm the intuition of Referee #1**: In Figure R2, we levitate a nanoparticle of 75 nm in radius to which we repeatedly apply our optimization routine 8 times in a row—each time initiating the procedure from a uniform wavefront. The routine is implemented using the same set of polynomials as in Figure 1, which is composed of 20 symmetric and 10 asymmetric components—and whose patterns are displayed in Figure S6. Along the *z* direction, we recorded stiffness gains ranging from 1.37 to 1.65. As an illustration, the results of three of these are provided in Figure R2a-c. Specifically, Figure R2a-c provides the projections of the three optimized wavefronts over the polynomial sets (“Coefficients”). The vertical dashed lines set the separations between symmetric and asymmetric distributions (i.e., polynomials), while their corresponding symmetric and asymmetric contributions (i.e., patterns) to the final wavefronts are provided in the insets. **Despite reaching similar stiffness enhancements, we clearly observe that the symmetric contributions differ substantially, while the asymmetric display similarities**. To be*

more quantitative, in Figure R2d and Figure R2e we perform, respectively, the correlations between the 8 symmetric and asymmetric contributions. We report mean correlations of about 40% for symmetric patterns and around 75% for asymmetric ones. **In short, we observe experimentally a certain “consistency” over the asymmetric patterns that are provided by the optimization, while more pronounced “variations” are reported over the symmetric ones.**

In light of these results, the intuition of Referee #1 appears correct. Regarding the symmetric patterns, we observe numerically that the stiffness optimization is characterized by multiple local minima. Thus, the fact that experimentally the symmetric patterns converge towards diverse solutions seems to indicate that, indeed, they **primarily increase the trap stiffness**. Concerning the asymmetric patterns, the larger degree of correlation between solutions seems to indicate that the optimization converges towards a final solution, in which certain **asymmetric polynomials systematically address the same “errors” that may correspond to “experimental inaccuracies”** (e.g., misalignments). Here, one can take as an example the contributions of Z_3^1 , Z_3^3 or Z_4^2 in Figure R2a-c, which seem to consistently emerge with large amplitudes to address specific optical aberrations. Yet, our results also show that on top of such systematic corrections, some **asymmetric polynomials can display strong variations that could be indicative (like in the symmetric case) of an effort to improve the stiffness**. Here, one can take as an example the contributions of Z_1^1 or Z_4^{-2} in Figure R2a-c, whose coefficients can flip sign to provide opposed contributions to the final wavefront.

Figure R2 : **Symmetric and asymmetric Zernike polynomials.** **a**, **b** and **c**, Expansion along the Zernike basis, Z_p^q , of three different wavefronts obtained by repeatedly running our optimization on the same nanoparticle (75 nm in radius). The vertical dashed line splits the axis symmetric polynomials and the asymmetric ones, while the symmetric and asymmetric components to the optimized wavefronts are provided in the inset. **d** (respectively **e**), Correlations between the symmetric (respectively asymmetric) distributions obtained after running the optimization of **a-c** 8 times in a row.

Related modification of the manuscript:

[1] *Supplementary Information: We wrote the new section 2.2 to discuss the contributions of symmetric and asymmetric polynomials. Figure R2 was added as the new figure Figure S7.*

[2] *Main text (page 4):*

“The wavefronts are expanded over a Zernike-polynomial basis, in which axis-symmetric and asymmetric elements appear to have different contributions (Supplementary Figure S5 and Figure R2)”.

In the case of axial Zernike polynomials ($m=0$) some connection to the theoretical analysis in SI 2.1 should be made, i.e. does Z^0_{38} resolve the narrowest rings considered in the optical force calculations? I can even think that one may perform the analysis based on the Fisher information how strongly each Zernike polynomial contributes to the final result (for future work).

*Following this suggestion by Referee #1, we used our numerical model to estimate the highest order of Zernike polynomials that should effectively have a substantial influence on our optimization. In Figure R3, we run the optimization while progressively increasing the number of rings that are used to discretize the optical phase. For a 5-ring discretization, the optimization is only partial and the optimal solution is reached above 15 rings. As a result, Zernike polynomials exhibiting a faster modulation (more rings) are likely to be irrelevant to the optimization. From Figure S6, one can conclude that the **symmetric polynomials Z^0_{32} to Z^0_{38} have probably close to no contribution to the optimization process.***

We also thank Referee #1 for mentioning the Fisher information, which could indeed be very useful for future work on this issue.

Figure R3 : Numerical optimization performed for different spatial resolutions. Evolution of the stiffness enhancement, $\kappa_{opt,z} / \kappa_{0,z}$, of a nanoparticle (125 nm in radius) when the spatial resolution (i.e., number of rings used to discretize the wavefront) is varied.

Related modification of the manuscript:

[1] *Supplementary Information: We wrote the new section 3.3 and Figure R3 was added as the new Figure S19a.*

Minor comments:

- 1- Page 2, sentence “However, this strategy is commonly believed to be ...” Is there any reference for this statement?

Here, we reformulated our sentence to avoid any ambiguity.

Related modification of the manuscript:

[1] Main text (page 2):

“However, this strategy **might reveal** ineffective for nanoparticles levitated in vacuum, which, unlike microparticles trapped in water, are expected to behave as point-like Rayleigh scatterers, displaying underdamped motion on very short timescales.”

- 2- Page 6, sentence “Our model also highlights that, even if the nanoparticle behaves as a Rayleigh scatterer (Mie parameter $2\pi R/\lambda \approx 0.74$) ...” I think that it quite established that for this size parameter the particle is not a Rayleigh scatterer. Actually, according to Wiscombe Applied Optics 1980 one should use up to 5th order multipole expansion in order to correctly describe scattering of such a particle size.

*We thank Referee #1 for highlighting this wording that might be misleading. Here, we intended that **our particle behaves as a “Rayleigh scatterer” in the sense that its electric dipole dominates over other contributions—as emphasized in Figure S9**. This is to be understood in opposition to a Mie particle for which multipoles can be extremely important. We agree with the reviewer that the work of Wiscombe [5] suggests in our case an expansion up to order $n=5$ of the scattered field on the Vectorial Spherical Harmonics (VSH). However, regarding optical forces, we observed in practice that numerical convergence is achieved well below this order. Specifically, an expansion to $n=2$ already provides an interesting convergence. This is emphasized in Figure R4, which displays the force landscapes (computed with an optimized SLM pattern), for several values of the VSH expansion order n . These simulations are consistent with a former work [6], which demonstrated that, under plane-wave excitation, an expansion up to second multipolar order (i.e., electric and magnetic dipoles) is sufficient to describe the forces exerted on a silica sphere of parameter 0.75.*

Related modification of the manuscript:

[1] Main text (page 6):

Our model also highlights that, even if the nanoparticle is **electric-dipole dominated** (Mie parameter $2\pi R/\lambda \approx 0.74$) [...].

Figure R4 : **Multipole expansion.** Longitudinal force profile computed for an optimized SLM phase mask, and using different cut-off orders, n , for the VSH expansion (see legend). The inset provides a zoomed-in view around the equilibrium point. The estimation provided for $n=2$ seems to display sufficient accuracy.

- 3- Figure 2c. The illustration is a bit misleading in a sense that the equilibrium position is not in the minimum of potential. I completely understand that only the gradient force potential is plotted but I would recommend adding a curve corresponding to scattering “potential.” The depiction would become cleaner.

Following Referee #1’s excellent suggestion, we now plot in Figure R5 in black the uniform potential, $U_0(z)$, from which we subtract the “work of the scattering force”, $\int_0^z F_{s,0}(u)du$. This “effective” potential reaches its minimum at the equilibrium location, $z_{eq} = 2.06 \mu m$, as already reported in the Figure 2a of the main text. Then, we plot in red the optimized potential, $U_{opt}(z)$, to which we subtract the “work of the scattering force”, $\int_0^z F_{s,opt}(u)du$. This “effective” potential reaches its minimum at the equilibrium location, $z_{eq} = 1.63 \mu m$ as reported in the Figure 2b of the main text.

Figure R5 : **Effective potentials (numerical results).** For the uniform (respectively optimized) wavefront used in Figure 2d of the main text, the black (respectively red) curve displays the optical potential $U_0(z)$ (respectively $U_{opt}(z)$) from which the work of the scattering force $F_{s,0}$ (respectively $F_{s,opt}$) is subtracted.

The minima $z_{eq,0}$ and $z_{eq,opt}$ mark the equilibrium positions when the uniform and the optimized wavefront are applied, respectively.

Related modification:

[1] *Supplementary Information: We included Figure R5 as Figure S13b and added a paragraph in section 3.2.*

[2] *Main text (page 6):*

“Alternatively, this stiffness improvement also reveals when considering the “effective” potential $U(z) - \int_0^z F_s(u)du$, whose confinement is substantially enhanced when the optimized profile is applied (Figure R5).”

- 4- Figure 2 caption. Reference to supplementary figure S7. Is it indeed S7 or probably S10?

Here the reference seems to be correct.

- 5- Page 8, sentence “... whose extension reveals much smaller than the particle size [16,24–27]” I think that a part of sentence is missing

We changed the sentence on page 8 from:

“Due to their non-conservative nature, scattering forces generate toroidal stochastic motions, whose extension reveals much smaller than the particle size [16,24–27].”

to

“Due to their non-conservative nature, scattering forces generate toroidal stochastic motions, which extend over distances much smaller than the particle’s radius [16,24–27].”

- 6- Discussion on Page 9: is the trapping frequency or stiffness enhancement influenced by the changing pressure? I mean can the tiny changes of refractive index decrease the trap strength enhancement? Or could any changes to vacuum chamber induce an extra phase shift that would influence the enhancement.

When pressure is reduced, the resonance frequency along each axis tends to slightly drop. This decrease originates from Duffing nonlinearities, which increase when gas damping is reduced. This behavior can be observed in Figure R6, where we plot the evolution of the resonances along all three axes when pressure is varied. We use both the uniform (dashed black) and optimized (red) profile used in Figure 4 of the main text. In particular, one can see that the longitudinal direction z (along which optical confinement is weaker and therefore nonlinearities less pronounced) is barely affected by the drop in pressure. In contrast, the transverse directions x and y (more subject to nonlinearities) fluctuate slightly more. Yet, despite these small changes, **the stiffness enhancement is not affected by the surrounding pressure.**

Figure R6 : **Frequencies as a function of pressure.** When pressure is varied in the vacuum chamber, the black (respectively red) curves display the evolution of the resonances along x (top), y (middle) and z (bottom) under the uniform (respectively optimized) beam profile used in Figure 4 of the main text.

Related modification:

[1] Supplementary Information: We included Figure R6 as the new Figure S16 and wrote the section 5.3.

[2] Main text (page 9):

“We show in the Figure R6 that central frequencies in the uniform and optimized case are barely reduced when pressure drops.”

7- Figure 4c, caption. It looks to me that σ_f is lower in the whole range of pressures, not just in the gray region

First, we would like to stress that we edited Figure 4 of main text to include measurement uncertainties (Figure R7). Then, as explained in reference [7], the evolution of the spectral linewidth is governed by a trade-off between two mechanisms: At relatively high pressure, the linewidth of the resonance is only set by the gas surrounding the nanoparticle and linearly evolves with pressure. When the pressure reaches a certain value, this linewidth becomes comparable with the broadening due to Duffing nonlinearities, which increase when pressure drops. This is why we constantly observe a linewidth reduction above ≈ 1 mbar and an improvement below. Regarding Referee #1's comment, above 1 mbar (i.e., outside of the grey regions), the linewidth is only set by the gas and is thus the same for both the optimized and uniform case in Figure R7c.

Figure R7 : Edited version of Figure 4 of main text.

Related modification:

[1] Main text (page 10): We inserted the edited version of Figure 4c and added a sentence in the caption

“Above ≈ 2 mbar, both curves display the same standard deviations.”

- 8- Supplementary Information: Please check the order of figures. Also correct the usage of comas around equations – on multiple occasions the paragraph after an equation starts with coma.

We would like to stress that in the Supplementary Material, we voluntarily use a numbering scheme where figures are ordered as they are referred to in the main text. The order of the figures as well as the comma issues have been checked and corrected.

- 9- SI section 1.3. Could you provide more details about implementation of the optimization procedure. I.e. for given set of Zernike polynomials a particle trajectory of what length is measured (points, time), how is the trajectory processed to obtain the resonant frequency. What is the total duration of the optimization procedure. How many times is typically trajectory recorded (iteration count in fminsearch does not mean number of function calls). Could you also comment on the stability of the trapping setup during the optimization?

Following Referee #1's suggestion, we wrote a paragraph in section 1.3 to address all these different aspects. Also, we added a diagrammatic description of our procedure that is provided in Figure R8 below.

Figure R8 : Diagrammatic description of the optimization process. At each iteration a uniform pattern is applied, a time trace is acquired, the PSDs are calculated and ultimately fitted to extract the

resonances $f_{x,y,z,0}$. The operation is repeated until being successful (overcoming, e.g., that the fitting does not converge). Then the modulated pattern is progressively applied onto the SLM, a time trace is acquired, the PSDs are calculated and ultimately fitted to extract the resonances $f_{x,y,z,opt}$. This operation is also repeated until being successful. The pattern is progressively removed and the cost function is evaluated.

Related modification:

[1] Supplementary Information: We included Figure R8 as the new Figure S18 and wrote a paragraph in the section 1.3.

[2] Methods:

“A complete description of the procedure is provided in Figure R8.”

10- SI page 4, last paragraph: is X a vector? If so please use bold font

The correction has been implemented.

11- Figure S3d. I would find useful to visually separate the axisymmetric coefficient from the nonsymmetric ones (by a space or different background)

We replaced the former figure mentioned by Referee #2 by Figure R2 that we introduced above in comment #2. This new figure is replicated as Figure R9 below and displays the separation suggested by Referee #2.

Figure R9 : Edited version of Figure S3.

12- SI page 7, sentence “Nonetheless, the optimization is less efficient than when performed directly on the particle of the proper dimension.” Please use size or radius instead of dimension.

The sentence has been modified accordingly.

13- Figure S4: I would suggest to add data from figure 1 as well for direct comparison.

Following Referee #1’s suggestion, we plot in Figure R10 the edited version of Figure S4.

Figure R10 : Edited version of Figure S4.

14- Figure S7ab. I would suggest to include dashed boxes that would show a magnified view in Fig. S7cd

Following Referee #1’s suggestion, we plot in Figure R11 the edited version of former Figure S7.

Figure R11 : Edited version of former Figure S7

15- SI page 9 paragraph 1. I think that it should be explicitly mentioned if the phase in each ring was constant or it also varied with polar angle which would probably introduce an equivalent of non-symmetric Zernike polynomials.

We edited the related paragraph in section 3.1 to stress that the phase is uniform in each ring.

16- SI page 9 sentence “Zoomed-in views of both landscapes are provided in Figs. S7a and d.” Probably Figs 7c and d.

The sentence was corrected.

17- SI page 7 sentence “We also note that the coordinate z in the optimized case has been shifted such that the conservative part of the force (i.e., gradient force, see section 2.2) remains zero at $z = 0$ (i.e., the beam focus).” I think that this sentence may be a bit hard to understand and I would recommend reformulating it.

The sentence was edited to:

“In the optimized case, the focal point is shifted along the z direction. To take that modification into account and locate the new focal point, the z coordinate has been shifted such that the conservative part of the force (i.e., gradient force $F_{g,opt}$, see section 2.2) is zero when the new coordinate reaches 0.”

18- SI Equation (5): define H

The correction has been implemented.

Reviewer #2 (Remarks to the Author):

Kleine et al study the use of wavefront shaping to improve the mechanical properties of silica nanoparticles optically trapped in vacuum. A spatial light modulator (SLM) is used to generate trapping beams with tailored spatial phase profiles resulting in enhanced mechanical spring constants when compared to a diffraction-limited trapping potential. The authors then use numerical simulations to attribute this effect to the suppression of non-conservative optical forces, which shifts the equilibrium position of the nanoparticle closer to the trap focus. Finally, they report on the reduction of nonlinearities in the mechanical motion of the nanoparticle at lower pressures.

These results **address key challenges in the fast-growing field of levitated optomechanics**. Ground-state cooling of levitated mechanical systems has only been achieved in optical traps, where laser light is known to generate unwanted motional decoherence via scattering and absorption of photons. **Wavefront shaping as demonstrated here could provide much-needed new tools to enhance trapping and suppress decoherence for quantum experiments with optically levitated nanoparticles**. The paper also shows that nonlinearities can be suppressed using wavefront shaping, which could **improve the stability of levitated oscillators in vacuum, another common problem with optical levitation**.

The main claim of the paper, that wavefront shaping can enhance optical trapping of levitated nanoparticles, **is justified and backed by experimental data** (Fig 1). **The paper is concise, well written and accessible to a broad readership**. Prior work is appropriately acknowledged. **I appreciate the careful analysis in the supplementary sections and the numerical simulations which provide detailed insights into the effect of tailored potential on the mechanics**.

However, the experimental plots lack error analysis (or mention of) throughout and there no strong theory plots to compare with the experimental datapoints. This makes all results (except Fig 1) less convincing. This work presents the novel experimental tool of wavefront shaping for optomechanics with levitated nanoparticles, **which has the potential to overcome key bottlenecks in the field**. **I believe is a worthwhile advance of relevance to the levitodynamics community**. I would recommend publications in Nature Communications, but only if the following comments are addressed:

We are very grateful to Referee #2 for this positive review and for acknowledging the impact our work could have onto the field of optical levitation. In what comes next, we provide a point-by-point response to the comments.

- 1- Based on Fig 1d, the authors state “This emphasizes that the optimization is more efficient on objects that are subject to stronger scattering forces”, indicating that their optimization works better with larger objects. Would this be the case even beyond the Rayleigh scattering regime? It would be useful to model the curves in Fig 1d to indicate the size range in which their optimization is beneficial and the maximum enhancement ratios that could be theoretically achieved.

To provide some context, as explained in Figure 2c of the main text, our mechanism relies on the fact that we are able to reduce the contribution of the scattering forces, while keeping the gradient force almost unchanged. In short, the optimization relies on the ability to reduce the role of the scattering force that tends to move the particle away from focus. In that sense, the stronger the scattering force, the better our approach operates. Thus, since larger particles display larger scattering cross section, they provide stronger scattering force to be improved and the optimization should work even better. At this point, we need to stress that the size of the particle one can levitate with a given optical setup is limited by the numerical aperture of the objective (NA). Typically, one can levitate nanoparticles up to a few hundreds of nm using NA from .7 to .8. To levitate larger particles (e.g., micronsize particles [3,8]) one must rely on smaller NA.

That being said, following Referee #2’s suggestion, we provide in Figure R12 new numerical simulations mimicking our experimental setup. Specifically, we run the optimization on a silica nanoparticle while varying its radius. As stated above, the optimization clearly improves as the size (and thus the scattering cross section) increases. In our current optical setup, these simulations show that trapping becomes instable when the radius exceed 140 nm, while an enhancement of almost 6 is predicted for particles of radius 135 nm. Nonetheless, we are convinced that this technique can be straightforwardly transferred to any levitation setup and, therefore, could serve to optimize the trapping of micronsize particles beyond the Rayleigh regime.

Figure R12 : Stiffness enhancement for various nanospheres’ radii (numerical results). The numerical model used in Figure 2 of the main text is adapted to optimize the stiffness ratio $\kappa_{opt,z}/\kappa_{0,z}$ while varying the radius of the silica nanoparticle. The enhancement improves as the radius of the particle—and therefore its scattering cross section—increases. For radii larger than 140 nm, the trap becomes unstable.

Related modification:

[1] Supplementary Information: Figure R12 was added as the new Figure S19b and we wrote the new section 3.3.

- 2- The authors refer to Fig S8 (numerical simulation) when claiming that “the optimized wavefront can produce the same value of k_z as in the uniform case with half the intensity”. This should be straightforward to test experimentally and could strengthen the results by showing agreement with theory. Why has this not been done?

*We thank Referee #2 for this suggestion. Indeed, in the former version of the text, we only relied on numerical simulations (see Figure S11) to claim that our optimization could serve to make trapping more photon-efficient. In Figure R13, we now provide the experimental results of an optimization performed on a nanoparticle of 125 nm in radius. This optimization was performed initially for a power emitted by the laser of 590 mW. Afterwards, the power was varied to confirm that the **stiffness enhancement is indeed independent of the laser power**. Owing to losses along the optical setup, the laser power does not correspond to the actual power focused onto the particle that sets closer to 100 mW.*

*When the lasing power is varied, Figure R13a displays in green, yellow and blue the enhancements that are measured along x , y and z respectively, while the vertical red dashed line marks the power P_{opt} at which the optimization was performed. Here, stiffness enhancements of $\kappa_{opt,x}/\kappa_{0,x} \approx 1.35$, $\kappa_{opt,y}/\kappa_{0,y} \approx 1.4$ and $\kappa_{opt,z}/\kappa_{0,z} \approx 2$ are reported regardless of optical power. The larger imprecision along z originates from the fact that the mechanical resonance along that direction is broad spectrally (lower optical confinement and quality factor). This induces a certain imprecision in the tracking of the resonance frequency f_z , which translates into larger fluctuations in the estimation of the stiffness enhancement $\kappa_{opt,z}/\kappa_{0,z}$. **This panel clearly emphasizes that the optimization is perfectly independent of the optical power.***

*Now, to specifically address Referee #2's question, in Figure R13b we focus on the stiffness enhancement of ≈ 2 that is achieved along z . **We expect that this profile will provide the same frequency as in the uniform case but with twice less power.** We plot in thick black and thick red, the mechanical resonances S_{zz} that are measured, respectively, under a uniform and an optimized wavefront when the optical power is set to 767 mW. We plot in light dark and light red, the mechanical resonances S_{zz} that are measured, respectively, under a uniform and an optimized wavefront when the optical power is set to 447 mW. For experimental reasons, we only achieved here a reduction of lasing intensity by a factor 1.7, which corresponds to the optimized resonance moving from 47 kHz down to 34.5 kHz (i.e., a reduction by a factor of $\approx \sqrt{1.8}$). Yet, we see that the optimized resonance for 447 mW (34.5 kHz) almost matches the uniform one obtained for 767 mW of power (32 kHz). Simple calculations confirm that both curves would align perfectly when using $767/2 = 383$ mW. **Therefore, we experimentally confirm that power can be substantially reduced with our method.***

Figure R13 : Variation of optical power. **a**, A 125 nm nanoparticle is optimized using a power emitted by the laser of 590 mW (vertical red dashed line, P_{opt}). Afterwards the laser power is varied from 447 mW to 767 mW. The enhancements $\kappa_{opt,x}/\kappa_{0,x}$, $\kappa_{opt,y}/\kappa_{0,y}$ and $\kappa_{opt,z}/\kappa_{0,z}$ are displayed in green, yellow and blue, respectively, and appear independent of optical power. **b**, For the optimization performed in **a**, the thick and light black (respectively red) curves display the mechanical spectra, S_{zz} , obtained for a uniform (respectively optimized) beam profile and for laser powers of 447 mW and 767 mW, respectively.

Related modification:

[1] *Supplementary Information: We added Figure R13 as the new Figure S10 and wrote the new section 2.3.*

[2] *Main text (page 6):*

“In line with experimental observations, simulations show that the trap is also more photon-efficient as the optimized wavefront can produce the same value of κ_z as in the uniform case but with now half the intensity (Figure R13 and Supplementary Figure S11)”.

- 3- Error bars are missing for all experimental plots (Fig 1d, Fig3c, Fig 4c), and no discussion of errors are provided. The strength of the conclusions is hard to judge without quantitative error bars.

We thank Referee #2 for this statement and acknowledge that such a discussion is needed. To address this comment, we will here consider individually the three experimental plots mentioned above (Fig 1d, Fig3c, Fig 4c) as errors bars do not have the same meaning each time.

We begin our discussion by considering Figure 1d of the main text, whose edited version is provided in Figure R14. Here, we recall that **we run a single time the optimization procedure on 4 different particles with different radii**. Each time the optimization converged towards final enhancements along x , y and z that are respectively plotted in green, yellow and blue. The corresponding optimized beam patterns are displayed in insets. **The difficulty in defining error bars goes back to the fact that our optimization process possesses multiple local minima that will lead to different wavefronts and enhancements**. For instance, for this reply we acquired new data for which we optimized 8 times in a row the same nanoparticle of 75 nm in radius. For three different optimizations, in Figure R15 we plot the expansion of the optimized wavefronts onto the Zernike basis used to encode the phase profile. There, we use a vertical dashed line to separate symmetric (left) and asymmetric vectors (right). In each panel, the symmetric and asymmetric components of the final patterns are displayed in the inset. These three examples clearly stress that the optimization converges towards different solutions, which provide different enhancements. Over the 8 optimizations that we have performed, the stiffness enhancements along x , y and z oscillate within the ranges [1.13,1.32], [1.17,1.34] and [1.35,1.65], respectively. **Thus, since the routine does not systematically converge towards the same solution the notion of error bar is hard to define**. For that reason, in the new version of Figure 1d (see Figure R14), where the routine converges towards a single solution for each radius, we define the error bars with respect to the fluctuations within the final configurations. In other words, after each optimization has converged to its final wavefront (provided in the inset), we measure the variation in stiffness enhancement when this pattern is applied multiple times. As such, the error bars correspond to the precision associated with a convergence towards a given minimum.

Figure R14 : Edited version of Figure 1d.

Figure R15 : **Symmetric and asymmetric Zernike polynomials.** **a**, **b** and **c**, Expansion along the Zernike basis, Z_p^q , of three different wavefronts obtained by repeatedly running our optimization the same nanoparticle (75 nm in radius). The vertical dashed line splits the axis symmetric polynomials and the asymmetric ones, while the symmetric and asymmetric components to the optimized wavefronts are provided in inset.

Related modification:

[1] *Supplementary Information: We discussed local minima in the new section 2.5 and added Figure R15 as the new Figure S7.*

[2] *Main text (page 5): We included the edited Figure 1d and added a sentence in the caption.*

Next, we consider Figure 3c of the main text, whose edited version is provided in Figure R16. This figure displays the averaged flux variance, $\langle J_v^2 \rangle$, of currents. Such currents are extracted from probability distributions in the velocity space (v_ρ, v_z) that are assembled out of time traces. To estimate the precision of this measurement at each pressure, we truncated our time traces into 30 pieces for which we extracted the corresponding $\langle J_v^2 \rangle$. Thus, by computing the standard deviation of this set of averaged flux variances, we are able to estimate the precision of our measurements. Interestingly, we observe that the measurement is very precise at high pressure and degrades for pressures below ≈ 1 mbar, which corresponds to the emergence of stochastic nonlinearities, which complexify the nanoparticles' dynamics.

Figure R16 : Edited version of Figure 3c.

Related modification:

[1] Main text (page 8): We included the edited Figure 3c and added a sentence in the caption.

At last, we consider Figure 4c of the main text, whose edited version is provided in Figure R17. These plots are assembled by looking at the histograms of resonances frequencies collected over 20 s acquisitions time traces. As explained in the main text, 5 ms-long time traces display narrow mechanical resonances that spectrally move over time. Here, σ_f relates to the width of these histograms that reduces when pressure is reduced down to 1 mbar and increases again below 1 mbar due to Duffing nonlinearities. To estimate the precision of such measurements, the 20 s time traces are split into 20 shorter traces providing 20 histograms. By systematically computing the width of these histograms, one obtains the error bars that are reported in Figure R17. Like in Figure R16 we observe that Duffing nonlinearities provide some complexity in the nanoparticles' dynamics, which ultimately degrades the precision.

Figure R17 : Edited version of Figure 4c.

Related modification:

[1] Main text (page 10): We included the edited Figure 4c and added a sentence in the caption.

- 4- Fig 4c indicates that the nonlinearity is suppressed below 1 mbar. However, the suppression seems to get worse again below 0.01mbar across all three dimensions. Why does this happen? Given the simulations of the trapping potential and scattering forces, could a theoretical comparison be provided for the curves in 4c? It would be useful to know (even theoretically) if suppression can be achieved also at lower pressures commonly needed for most experiments in the field, particularly those motivated by quantum applications.

We thank Referee #2 for this suggestion. First, we would like to mention that the introduction of error bars in Figure R17 (as suggested by Referee #2 above) emphasizes that the precision on the nonlinear broadening σ_f worsens at low frequencies. Therefore, we cannot be 100% confident that the reduction of nonlinearities effectively degrades below .01 mbar. For instance, Figure S17b seems to indicate that the reduction is preserved at low pressure along the transverse directions x and y . Moreover, owing to the stochastic nature of thermal fluctuations, investigating numerically the evolution of nonlinear effects with pressure would require the development of a whole new numerical tool package, which, unfortunately, goes beyond the scope of the present work.

Yet, a theoretical discussion regarding the mechanisms at play can be nevertheless possible by considering the work by Gieseler et al. [7]. The motion of a levitated nanoparticle of mass m along a direction $q_i \in [x, y, z]$ fulfills:

$$\ddot{q}_i + \Gamma_i \dot{q}_i + \Omega_i^2 \left(1 - \sum_{j \in [x, y, z]} \xi_j q_j^2 \right) q_i = \frac{F_{fl}(t)}{m}, \quad (1)$$

in which Ω_i , Γ_i and ξ_j stand for the mechanical frequency, the damping of the surrounding gas and Duffing nonlinearities, respectively. In equation (1), $F_{fl}(t)$ stands for a Langevin force related to Γ_i through the fluctuation-dissipation theorem. The damping provided by the surrounding gas is proportional to its pressure P and fulfills:

$$\Gamma_i = \frac{64r_i^2}{3m\bar{v}} P \quad (2)$$

where r_i and \bar{v} correspond to the nanoparticle's dimension along q_i and the averaged velocity of the gas' molecules, respectively. For the rest of this discussion, we drop the subscript "i" for simplicity. In their work, Gieseler et al. explained that the evolution of the linewidth as a function of pressure is governed by two competing effects, which we will respectively refer to as linear, $\Delta\Omega_L$, and nonlinear broadening, $\Delta\Omega_{NL}$. Set by gas damping, linear broadening reads $\Delta\Omega_L = \Gamma$ and is therefore proportional with the pressure. Resulting from Duffing nonlinearities, nonlinear broadening reads $\Delta\Omega_{NL} = \frac{3\xi}{4\Omega} k_B T$, in which T stands for the internal temperature of the particle. At high pressures, $\Delta\Omega_L$ dominates over $\Delta\Omega_{NL}$. Close to ≈ 1 mbar, $\Delta\Omega_L$ becomes smaller than $\Delta\Omega_{NL}$, which then becomes the dominant broadening mechanism. Below ≈ 1 mbar, $\Delta\Omega_{NL}$ increases when pressure is reduced owing to a combination of factors, like the augmentation of the nanoparticle's internal temperature T at low pressure and a strengthening of nonlinear effects [9]. The competition between both broadening mechanisms is clearly reported in Figure R17.

As explained in Figure 2c of the main text, when the optimization is performed, the particle is brought closer to focus, which increases the mechanical frequency, Ω , but also reduces the nonlinearities, ξ . This point is emphasized in the numerical simulation displayed in Figure R18.

There we plot for a uniform (black) and an optimized wavefront (red) the optical potential, $U(z)$, from which we subtract the “work of the scattering force”, $\int_0^z F_s(u)du$. In the optimized case, the confinement is stiffer (i.e., larger Ω) and more quadratic (i.e., reduced Duffing nonlinearity ξ). As a result, the optimization strongly reduces the ratio $\frac{\xi}{\Omega}$ that governs the evolution of $\Delta\Omega_{NL}$. Thus, when nonlinear broadening becomes the dominant mechanism, evolution with pressure is substantially reduced. **From this reasoning, one can indeed expect that nonlinearities are reduced by optimization at even lower pressure than what is displayed in Figure R17 or in Figure S17.** Yet, the model provided by Gieseler et al. is hard to extend to very low pressure at which nonlinearities become more pronounced and new mechanism might come into play.

Figure R18 : **Effective potentials (numerical results)**. For the uniform (respectively optimized) wavefront used in Figure 2d of the main text, the black (respectively red) curve displays the optical potential $U_0(z)$ (respectively $U_{opt}(z)$) from which the work of the scattering force $F_{s,0}$ (respectively $F_{s,opt}$) is subtracted. The minima $z_{eq,0}$ and $z_{eq,opt}$ mark the equilibrium positions when the uniform and the optimized wavefront are applied, respectively.

Related modification:

[1] *Supplementary Information: We wrote the new sections 5.1 and 5.2 to discuss the reduction of nonlinear broadening with pressure. We also added Figure R18 as the new Figure S13b and wrote a dedicated paragraph in section 3.2.*

- 5- A few suggestions on terminology:
 “the stiffness of the oscillator that governs its dynamics” – “the spring constant (stiffness) that governs the oscillator dynamics”
 “magnifications” – “enhancement factors”
 “whose extension reveals much smaller” – “whose extension is much smaller.”

We thank Referee #2 for these suggestions and implemented them throughout the text.

Reviewer #3 (Remarks to the Author):

Review of the manuscript: “Wavefront Shaping of Scattering Forces Enhances Optical Trapping of Levitated Nanoparticles” for publication in Nature Communications.

The manuscript by Kleine et al demonstrates a new iterative experimental method to enhance the optical trap confinement for dielectric spherical nanoparticles. It is based on SLM assisted wavefront shaping and consists in selective reduction of the scattering force while keeping the gradient force almost unaffected. As the result, the nanoparticle experiences a smaller axial shift away from the waist of the tightly focused trapping beam, which in turn leads to higher mechanical oscillation frequencies along all three directions with best enhancement achieved in the along the laser direction, corresponding to the lowest oscillation frequency. The reduction of the scattering force was demonstrated experimentally with the measurement of the joint probability distribution and current of the radial and axial particle velocities: in case of the optimized wavefront the magnitude of the vortex current in the phase space is smaller. Finally, the optimized confinement was used to show reduction of the thermal nonlinearities experienced by the particle at lower pressures thanks to a better confinement of the particle motion within the harmonic region of the trap.

The nanoparticles used in this work are mostly Rayleigh scatterers with small deviations towards Mie scattering. This work goes beyond state of the art by showing that wavefront shaping can indeed outperform the diffraction-limited focusing, which was thought optimal Rayleigh particles. Therefore, the claims of the paper are convincing and extensively explored. **The paper is written clearly and both experimental implementation and the experimental / numerical analysis are shown in a comprehensive way. The originality of the research approach is making use of wavefront shaping, commonly used for controlling light in complex media, for optimizing the control of nanoparticle motion. This will be of interest for the community exploring quantum optomechanics with trapped nanoparticles.**

I would therefore definitely recommend the publication of this manuscript in Nature Communication under condition that the authors address the following minor issues that I found reading this manuscript:

We are grateful to Referee #3 for this very positive review and provide a point-by-point response to the comments below.

- 1- For smaller nanoparticle radius the stiffness enhancement reduces. While the authors show that wavefront shaping mostly enhances the trapping due to the electric dipole force, I wonder whether the stiffness optimization would still work for perfect Rayleigh scatterers, for example single atoms in optical tweezers?

Actually, as we explain in Figure 2c of the main text, our optimization relies on the fact that we are able to reduce the contribution of the scattering force, while keeping the gradient contribution almost unchanged. In the case of our silica nanoparticle, the object can be modeled as a pure electric dipole when its radius becomes lower than ≈ 70 nm. Since we demonstrated experimentally the optimization of nanoparticles of 75 nm in radius, this convinces us that our method should be able to operate down to the Rayleigh regime.

Now, regarding atoms, as stated above our optimization relies on the ability to reduce the role of the scattering force that tends to move the particle away from focus. In that sense, the stronger the scattering force, the better our approach operates. The strength of the scattering force respectively to the gradient force is directly proportional to the ratio $Im(\alpha)/Re(\alpha)$, where α stands for the polarizability of the particle. Thus, for nanoparticles, this ratio increases with size and/or refractive index, which will improve the optimization's performances. In the case of atoms, the ratio becomes very large at or near resonance. As such atoms trapped using a near-resonant wavelength should greatly benefit from this approach.

Related modification:

[1] Supplementary Information: We added a discussion in the new section 3.2

- 2- The scattering force is usually associated with the scattering cross section of a particle. In standard textbooks the scattering cross sections are usually defined for incident plane wave illumination, while the situation may be much different in case of a spatial phase modulated beam. Does wavefront shaping actually modify the effective scattering cross section of the particle? If yes, could it then also be used to modulate the absorption cross section in case of absorbing particles?

As Referee #3 points out correctly, the scattering and the absorption cross section are quantities generally defined under plane wave illumination and depend only on the particle's properties. However, by redistributing the incident field, wavefront shaping can strongly modify the effective coupling to scattering or absorption channels, enhancing or suppressing these processes. In this sense, wavefront shaping can indeed be used to modulate the effective scattering or absorption experienced under structured illumination. A very interesting example in this context is the work by Hong et al. [10], in which wavefront shaping is used to modulate the scattering and absorption of metallic nanoparticles.

Related modification:

[1] Main text (page 11):

“For instance, it could help address major challenges in optical levitation, including stable trapping in high vacuum, reducing absorption losses and levitate particles composed of diverse materials [11], complex geometries [12] or embedded with emitters [13,14].”

- 3- As far as I know, Silica is a nonmagnetic material and in the SupMat Section 1.1 I haven't found any information of the magnetic properties of the nanoparticles allowing to calculate the magnetic dipole polarizability. However, on Fig 2 as well as Fig S7, I see that the MD contribution seems important. Could the authors please provide more details on the magnetic dipole polarizability α_{MD} , used in the calculations?

*When light is scattered by a particle much smaller than its wavelength, the electric field, the displacement vector and the electric current are all uniform quantities inside the particle. This induces only an electric dipole. When the size of this particle is progressively increased, the electric current patterns become more complex inside the particle, as the field begins to depart from a uniform value. Some of these current loops **generate magnetic moments even though the material itself is not magnetic** [15]. Thus, even though the particle is intrinsically dielectric, it can display magnetic multipoles (dipole, quadrupole, octupole... depending on its dimension). Here, we would like to mention a very interesting review on the matter [16].*

Related modification:

[1] Supplementary Information: We edited the first paragraph of section 3.1.

- 4- The scattering force can be expressed as the product of the laser intensity and the gradient of the optical phase (see for example the equation 14.44 in the "Nano Optics" book by Lukas Novotny). On Fig. S8 c one can clearly see the reduction of the laser intensity along z between 0 and 3 μm , while the slope of intensity is not changed. This in turn reduces the scattering force while keeping the gradient force unchanged. But is there any additional contribution from the phase? Does wavefront shaping also affect the beam's wavefront at the trapping area in a way which could reduce the gradient of the phase?

*When considering the case of an electric dipole (as in Eq.14.44 of "NanoOptics" referred to by Referee #3), the gradient force originates from the gradient of the optical intensity, while the scattering force originates from the gradient of phase (times the intensity). Therefore, the contribution of the phase is already and solely encoded in the scattering force. **In short, the optimization handles the phase completely and there is no additional contribution from the phase.***

- 5- The authors describe in details the procedure for the stiffness optimization in the SupMat Section 1.3, but I still miss the understanding of what is actually changed between each iteration: are the phase patterns corresponding to each Zernike modes tested one after the other at different contrasts?

Following Referee #3's suggestion, we wrote a paragraph in section 1.3 to address all these different aspects. Also, we added a diagrammatic description of our procedure that is provided in Figure R19 below.

Figure R19 : **Diagrammatic description of the optimization process.** At each iteration: A uniform pattern is applied, a time trace is acquired, the PSDs are calculated and ultimately fitted to extract the resonances $f_{x,y,z,0}$. The operation is repeated until being successful (e.g., if the fitting does not converge). Then modulated pattern is progressively applied onto the SLM, a time trace is acquired, the PSDs are calculated and ultimately fitted to extract the resonances $f_{x,y,z,opt}$. This operation is also repeated until being successful. The pattern is progressively removed and the cost function is evaluated.

Related modification:

[1] Supplementary Information: We included Figure R19 as Figure S18 and wrote a paragraph in the section 1.3.

[2] Methods:

“A complete description of the procedure is provided in Figure R8.”

- 6- Do the authors think it is possible to implement a non-iterative stiffness enhancement approach, similar to the Transmission Matrix measurement in static complex media [17], but based on the measurement of the particle's mechanical response to each applied mode?

We are convinced that such a method should be implementable using a non-iterative approach. For instance, we would like to bring to the attention of Referee #3 this paper by Mazilu et al. [18], in which they introduced a mathematical scheme allowing to optimize scattering interactions using an approach similar to the one developed by Popoff et al. [17].

Related modification:

[1] Main text (page 11):

“Building upon the remarkable progress recently achieved in wavefront shaping [19,20], we anticipate that this technique could be implemented non-iteratively [18] and transposed to different properties.”

- 7- The authors give the value of 430 nm for the beam waist which is roughly 0.404×1064 nm and the axial equilibrium position of $z_{eq} = 2$ μm for a $R = 125$ nm particle. These values seemed to me too small for the waist and too large (above the Rayleigh length) for the equilibrium position. By using the beam parameters: $NA = 0.8$ (Numerical Aperture of the trapping lens), $f_0 = 1.1$ (overfilling defined as the ratio between the Gaussian beam waist before the lens and the lens radius) and the particle size $R = 125$ nm, with help of the Optical Trapping Toolbox on MATLAB I found that a Gaussian waist $w_0 = 676$ nm (which is $\sqrt{0.404} \times 1064$ nm) fits best the radial beam profile along x for a x -polarized incident beam (I used the definition: $I(x) = I_0 \exp(-2x^2/w_0^2)$). I also got the axial equilibrium position of 0.8×1064 nm which is approximately 850 nm. Could the authors please verify / justify the values presented in the manuscript?

*Here, we would like to apologize to Referee #3 as this value is indeed incorrect. This is a typo from an older version that was not corrected before submission. Moreover, we thank Referee #3 for taking the time to run a simulation that confirm our mistake. **At last, we want to stress here that we mention this value in the Methods for indicative purposes and that it was not used at any stage in our work.** The waist is estimated to lie around 810 nm.*

- 8- On Fig. 3 c there are significant variations of the averaged flux variances but no errorbars. Would it be possible to discuss these variations or to provide estimated errorbars?

An edited version of Figure 3c of the main text is provided in Figure R20. This figure displays the averaged flux variance, $\langle J_v^2 \rangle$, of currents. These currents are extracted from probability distributions in the velocity space (v_ρ, v_z) that assembled out of time traces. To estimate the precision of this measurement at each pressure, we truncated our time traces into 30 pieces for which extract the corresponding $\langle J_v^2 \rangle$. Thus, by computing the standard deviation of this set of averaged flux variances, we are able to estimate the precision of our measurements. Interestingly, we observe that the measurement is very precise at high pressure and degrade when pressure below ≈ 1 mbar, which corresponds to the emergence of stochastic nonlinearities, which complexify the nanoparticles' dynamics.

Figure R20 : Edited version of former Figure 3c.

Related modification:

[1] Main text (page 8): We include the edited Figure 3c and add a sentence in the caption.

- 9- Just wanted to draw the attention of the authors to the fact that they give the value of 2200 kg/m³ for the mass density of Silica. It indeed agrees with the bulk density of Silica, but differs from the value of 1850 kg/m³ for the Silica nanoparticles provided by MicroParticles GmbH: <https://microparticles.de/en/properties>.

We thank Referee #3 for this remark. Actually, the density was chosen on purpose to be 2200 kg/m³ even though the particles in solution possess a density of 1850 kg/m³. The silica particles used in optical levitation are actually porous, thus explaining their density lower than bulk silica. When pressure is reduced, these pores collapse for pressure close to ≈ 1 mbar, which leads to a substantial increase of their density. A complete description of this issue is described in different papers such as reference [21].

Related modification:

[1] Supplementary Information: We added a sentence in section 1.1 to say that the nanoparticles are densified at low pressure.

Bibliography

- [1] M. Zheng, S. Chen, B. Liu, Z. Weng, and Z. Li, Fast measurement of the phase flicker of a digitally addressable LCoS-SLM, *Optik (Stuttg)* **242**, (2021).
- [2] K. H. Fan-Chiang, S. H. Huang, C. Y. Shen, H. L. Wang, Y. W. Li, H. C. Tsai, and Y. P. Huang, Analog LCOS SLM devices for AR display applications, *J Soc Inf Disp* **28**, 581 (2020).
- [3] F. Monteiro, S. Ghosh, A. G. Fine, and D. C. Moore, Optical levitation of 10-ng spheres with nano- μg acceleration sensitivity, *Phys Rev A (Coll Park)* **96**, 063841 (2017).
- [4] J. García-Márquez, V. López, A. González-Vega, and E. Noé, Flicker minimization in an LCoS spatial light modulator, *Opt Express* **20**, 8431 (2012).
- [5] W. J. Wiscombe, Improved Mie scattering algorithms, *Appl Opt* **19**, 1505 (1980).
- [6] M. Riccardi, A. Kiselev, K. Achouri, and O. J. F. Martin, Multipolar expansions for scattering and optical force calculations beyond the long wavelength approximation, *Phys Rev B* **106**, (2022).
- [7] J. Gieseler, L. Novotny, and R. Quidant, Thermal nonlinearities in a nanomechanical oscillator, *Nat Phys* **9**, 806 (2013).
- [8] F. Monteiro, W. Li, G. Afek, C.-L. Li, M. Mossman, and D. C. Moore, Force and acceleration sensing with optically levitated nanogram masses at microkelvin temperatures, *Phys Rev A (Coll Park)* **101**, 53835 (2020).
- [9] Y. Amarouchene, M. Mangeat, B. V. Montes, L. Ondic, T. Guérin, D. S. Dean, and Y. Louyer, Nonequilibrium Dynamics Induced by Scattering Forces for Optically Trapped Nanoparticles in Strongly Inertial Regimes, *Phys Rev Lett* **122**, 1 (2019).
- [10] P. Hong and W. L. Vos, Controlled light scattering of a single nanoparticle by wave-front shaping, *Phys Rev A (Coll Park)* **106**, (2022).
- [11] S. Kuhn, B. A. Stickler, A. Kosloff, F. Patolsky, K. Hornberger, M. Arndt, and J. Millen, Optically driven ultra-stable nanomechanical rotor, *Nat Commun* **8**, (2017).
- [12] L. Bellando, M. Kleine, Y. Amarouchene, M. Perrin, and Y. Louyer, Giant Diffusion of Nanomechanical Rotors in a Tilted Washboard Potential, *Phys Rev Lett* **129**, (2022).
- [13] T. Delord, P. Huillery, L. Nicolas, and G. Hétet, Spin-cooling of the motion of a trapped diamond, *Nature* **580**, 56 (2020).
- [14] Y. Jin, K. Shen, P. Ju, X. Gao, C. Zu, A. J. Grine, and T. Li, Quantum control and Berry phase of electron spins in rotating levitated diamonds in high vacuum, *Nat Commun* **15**, (2024).
- [15] A. B. Evlyukhin, S. M. Novikov, U. Zywietz, R. L. Eriksen, C. Reinhardt, S. I. Bozhevolnyi, and B. N. Chichkov, Demonstration of Magnetic Dipole Resonances of Dielectric Nanospheres in the Visible Region, *Nano Lett* **12**, 3749 (2012).
- [16] M. Riccardi, A. Kiselev, K. Achouri, and O. J. F. Martin, Multipolar expansions for scattering and optical force calculations beyond the long wavelength approximation, *Phys Rev B* **106**, (2022).
- [17] S. M. Popoff, G. Lerosey, R. Carminati, M. Fink, a. C. Boccara, and S. Gigan, Measuring the Transmission Matrix in Optics: An Approach to the Study and Control of Light Propagation in Disordered Media, *Phys Rev Lett* **104**, 100601 (2010).
- [18] M. Mazilu, J. Baumgartl, S. Kosmeier, and K. Dholakia, *Optical Eigenmodes; Exploiting the Quadratic Nature of the Energy Flux and of Scattering Interactions*, 2011.
- [19] S. Rotter and S. Gigan, Light fields in complex media: Mesoscopic scattering meets wave control, *Rev Mod Phys* **89**, 015005 (2017).

- [20] H. Cao, A. P. Mosk, and S. Rotter, Shaping the propagation of light in complex media, *Nat Phys* **18**, 994 (2022).
- [21] C. Li et al., Morphological Tracking and Tuning of Silica Nanoparticles in Optomechanical Systems for Enhanced Stable Levitation in Vacuum, *ACS Appl Nano Mater* **7**, 25493 (2024).

Wavefront Shaping of Scattering Forces Enhances Optical Trapping of Levitated Nanoparticles (NCOMMS-25-36074A)

First of all, we are very grateful to the three referees for their reviews. Regarding Referee #2 in particular, we would like to thank them for accepting the publication of our work. As it appears that there has been a small misunderstanding in our first reply, we clarify this point below.

Reviewer #2 (Remarks to the Author):

The authors have provided a comprehensive response to my queries. They have shown experimental verification that power can be reduced (by almost a factor of 2) to achieve the same stiffness (addressing query 2), added appropriate error bars to their figures (addressing query 3), and provided discussions showing that their techniques are useful even at lower pressures (addressing query 4).

I do believe they misunderstood my first query about the effectiveness of their techniques for particle radius beyond a couple hundred nm. They describe that for the smaller particles that are typically trapped with 0.7 or 0.8 NA lenses (as is the case in their experimental setup), their optimisation shows improvement with size.

However, the question was about larger particles that require lower NA trapping; does their optimisation still hold, and if not, is there an optimal size range where their techniques work? I do not think statements such as "these simulations show that trapping becomes unstable when the radius exceed 140nm" and "Nonetheless, we are convinced that this technique can be straightforwardly transferred" are convincing (where in the simulations does the instability arise when they could change the NA in the simulations, and what convinces them that their techniques can be transferred?).

Nevertheless, this is perhaps a minor issue that simply needs clarification for completeness and readability. Overall, based on their modifications in response to referee feedback, the paper is much stronger now and I am happy to recommend publication.

First, we would like to apologize for this misunderstanding.

In our first reply, we run new simulations to mimic our experimental setup (i.e., $NA = 0.8$), which limits the size of the particle that can be trapped. Yet, we totally agree with Referee #2 that if one assumes that the NA can be adjusted at will, the stability of the trap can always be satisfied.

Now, regarding the extension of our technique to larger particles, we are convinced that it should actually reveal even more efficient. Indeed, as the size of the particle increases, the object displays more electric/magnetic multipoles. Such multipoles can then be harnessed by wavefront shaping [1] to assemble very complex optical-field distributions across the particle. Therefore, we anticipate that it should lead to a greater controllability over the optical forces and ultimately stronger enhancements of the optical stiffness.

At last, we would like to inform Referee #2 that we are currently working on applying this technique to larger particles and hope to be able to propose soon a complete experimental and theoretical study on this matter.

Related modification:

[1] Supplementary Information: We edited section 3.3 to clarify the two statements mentioned by Referee #2:

“In our current optical setup (i.e., when considering a NA of 0.8), these simulations show that trapping becomes instable for radii exceeding 140 nm, while an enhancement of almost 6 is predicted for particles of radius 135 nm. We are also convinced that this technique can be straightforwardly transferred to optimize micronsized particles levitated using lower NAs. For larger particles, the presence of multiple electric and magnetic dipoles provides a deeper controllability over the light field [1], which can be harnessed to design strong optical confinements. As such, we anticipate that our approach should reveal even more efficient in the context of larger levitated objects.”

Bibliography

- [1] P. Hong and W. L. Vos, Controlled light scattering of a single nanoparticle by wave-front shaping, Phys Rev A (Coll Park) 106, (2022).